# Many si/shRNAs can kill cancer cells by targeting multiple survival genes through an off-target mechanism

**William Putzbach[1†], Quan Q Gao[1†], Monal Patel[1†], Stijn van Dongen[2], Ashley Haluck-Kangas[1], Aishe A Sarshad[3], Elizabeth T Bartom[4], Kwang-Youn A Kim[5], Denise M Scholtens[5], Markus Hafner[3], Jonathan C Zhao[1], Andrea E Murmann[1], Marcus E Peter[1,4]***

[1]Division of Hematology and Oncology, Department of Medicine, Northwestern University, Chicago, United States; [2]European Bioinformatics Institute (EMBL-EBI), Cambridge, United Kingdom; [3]Laboratory of Muscle Stem Cells and Gene Regulation, National Institute of Arthritis and Musculoskeletal and Skin Diseases, National Institutes of Health, Bethesda, United States; [4]Department of Biochemistry and Molecular Genetics, Northwestern University, Chicago, United States; [5]Department of Preventive Medicine, Feinberg School of Medicine, Northwestern University, Chicago, United States

**\*For correspondence:** m-peter@ northwestern.edu

[†]These authors contributed equally to this work

**Competing interests:** The authors declare that no competing interests exist.

**Abstract** Over 80% of multiple-tested siRNAs and shRNAs targeting CD95 or CD95 ligand (CD95L) induce a form of cell death characterized by simultaneous activation of multiple cell death pathways preferentially killing transformed and cancer stem cells. We now show these si/shRNAs kill cancer cells through canonical RNAi by targeting the 3'UTR of critical survival genes in a unique form of off-target effect we call DISE (death induced by survival gene elimination). Drosha and Dicer-deficient cells, devoid of most miRNAs, are hypersensitive to DISE, suggesting cellular miRNAs protect cells from this form of cell death. By testing 4666 shRNAs derived from the CD95 and CD95L mRNA sequences and an unrelated control gene, Venus, we have identified many toxic sequences - most of them located in the open reading frame of CD95L. We propose that specific toxic RNAi-active sequences present in the genome can kill cancer cells.
DOI: https://doi.org/10.7554/eLife.29702.001

## Introduction

One of the most popular methods utilized to reduce gene expression in cells is RNA interference (RNAi). RNAi has been used in several studies to identify genes critical for the survival of human cancer cell lines (*Cowley et al., 2014*; *Hadji et al., 2014*; *Hart et al., 2014*; *Morgens et al., 2016*; *Wang et al., 2015*). During RNAi, gene expression is inhibited by small interfering (si)RNAs, small hairpin (sh)RNAs or micro (mi)RNAs. miRNAs are generated as primary transcripts in the nucleus where they undergo processing to pre-miRNAs by the Drosha-DGCR8 complex before being exported to the cytosol by exportin 5 (*Ha and Kim, 2014*; *Krol et al., 2010*). Once in the cytosol, pre-miRNAs and shRNAs are cleaved by Dicer, a type III RNase that functions in complex with TRBP, generating 21–23 nucleotide long fragments of double-stranded RNA (dsRNA) that have two nucleotide 3' overhangs (*Zamore et al., 2000*). DsRNA fragments or chemically synthesized double-stranded siRNAs are loaded into the RNA-induced silencing complex (RISC) as single-stranded RNAs (the guide RNA) (*Siomi and Siomi, 2009*). A near-perfect complementarity between the guide strand of the si/miRNA and the target mRNA sequence results in cleavage of the mRNA (*Pratt and*

**eLife digest** Cells store their genetic code within molecules of DNA. Some of this information will be copied into chemically similar molecules called RNAs, from which the sequence of letters in the genetic code can be translated to build proteins. However, these messenger RNAs are not the only RNA molecules that cells can make. MicroRNAs are other short pieces of RNA that closely match sequences in parts of certain messenger RNAs. The messenger RNAs targeted by microRNAs are broken down inside the cell, which reduces how much protein can be produced from them. Since its discovery, scientists have exploited this process – called RNA interference (or RNAi for short) – and designed microRNA-like small interfering RNAs (siRNAs) to target particular messenger RNAs and decrease the levels of the corresponding proteins in countless experiments.

Two proteins that have been studied in RNAi experiments are CD95 and its interaction partner CD95L. Both of these proteins are important in human cancer cells, and targeting them via RNAi killed cancer cells in an unknown mechanism that the cancer cells were unable to resist.

RNAi experiments are designed to be specific, but sometimes they can accidently target other non-target messenger RNAs. Putzbach, Gao, Patel et al. have now analyzed all of the siRNAs that can be made from the messenger RNAs for CD95 and CD95L to mediate RNAi in cancer cells. This revealed that several messenger RNAs, other than those for CD95 and CD95L, were unintentionally being targeted, including many that code for proteins that cells need to survive. Further examination of the messenger RNA for CD95 and CD95L showed that they contain short sequences that are similar to those in the messenger RNAs of the genes that encode these survival proteins. Putzbach et al. were able to study and then predict which siRNA sequences would be toxic to cancer cells.

These findings indicate that an RNAi off-target effect may actually be used to kill cancer cells. Future studies will determine whether this effect could be exploited to shrink tumors in animal models of cancer. If successful, this in turn could lead to new treatments for cancer patients.

DOI: https://doi.org/10.7554/eLife.29702.002

MacRae, 2009). Incomplete complementarity results in inhibition of protein translation and contributes to mRNA degradation (Guo et al., 2010). mRNA targeting is mostly determined by the seed sequence, positions 2-7/8 of the guide strand, which is fully complementary to the seed match in the 3'UTR of targeted mRNAs. Similar to miRNAs, although not fully explored, siRNAs and shRNAs also target multiple mRNAs besides the mRNAs they were designed to silence—a phenomenon commonly referred to as off-target effect (OTE)—that is generally sought to be avoided (Birmingham et al., 2006; Jackson et al., 2006; Lin et al., 2005).

The death receptor CD95 (Fas/APO-1) mediates induction of apoptosis when bound by its cognate ligand CD95L, most prominently in the context of the immune system (Krammer, 2000). However, more recently, it has become apparent that the CD95/CD95L system has multiple tumor-promoting activities (Peter et al., 2007). CD95 signaling promotes cell growth (Chen et al., 2010), increases motility and invasiveness of cancer cells (Barnhart et al., 2004; Kleber et al., 2008), and promotes cancer stemness (Ceppi et al., 2014; Drachsler et al., 2016; Qadir et al., 2017). In fact, we reported tumors barely grew *in vivo* when the CD95 gene was deleted (Chen et al., 2010; Hadji et al., 2014). Therefore, it appeared consistent that multiple shRNAs and siRNAs targeting either CD95 or CD95L slowed down cancer cell growth (Chen et al., 2010) and engaged a distinct form of cell death characterized by the activation of multiple cell death pathways (Hadji et al., 2014). This unique form of cell death cannot be inhibited by conventional cell death or signaling pathway inhibitors or by knockdown of any single gene in the human genome (Hadji et al., 2014); it preferentially affects transformed cells (Hadji et al., 2014) including cancer stem cells (Ceppi et al., 2014). Here, we report that loading of CD95 and CD95L-derived sequences (si/shRNAs targeting CD95 or CD95L) into the RISC elicits a distinct form of cell death that results from the targeting of multiple survival genes in a unique form of OTE.

## Results

### si/shRNAs kill cells in the absence of the targeted site

More than 80% of multiple-tested shRNAs or siRNAs designed to target either CD95 or CD95L were toxic to multiple cancer cells (*Hadji et al., 2014*). We have now extended this analysis to Dicer substrate 27mer DsiRNAs designed to target CD95L (*Figure 1—figure supplement 1A*, [*Kim et al., 2005*]). All five DsiRNAs displayed toxicity when introduced into HeyA8 cells at 5 nM (*Figure 1—figure supplement 1B*) reinforcing our previous observation that the majority of CD95 and CD95L targeting si/shRNAs are toxic to cancer cells. We also analyzed a data set of a genome-wide analysis of 216 cells infected with a pooled library of the TRC shRNAs (*Cowley et al., 2014*). Most of the shRNAs we have tested were found to be depleted in the infected cell lines included in this study. The following shRNAs were found to be depleted in the listed percentage of the 216 cell lines tested: shL4 (99.5%), shL1 (96.8%), shR6 (88.9%), shR7 (75%), shR3 (71.8%), shL2 (67.1%), shR5 (38.4%), shL5 (26.4%), and shR8 (21.3%) (*Figure 1—figure supplement 1C*). Consistent with our data, shL1 and shR6 were found to be two of the most toxic shRNAs. Again in this independent analysis, the majority of tested shRNAs (67%) targeting either CD95 or CD95L killed more than half of all tested cancer cell lines.

Interestingly, a more recent RNAi screen did not report toxicity after expressing shRNAs against CD95 or CD95L (*Morgens et al., 2016*). The authors of this study used a second-generation shRNA platform based on a miR-30 backbone. To determine the source of the discrepancy in the data, we generated miR-30-based Tet-inducible versions of some of our most toxic shRNAs (shL1, shL3, shL4, shR5, shR6, and shR7, *Figure 1—figure supplement 2A*) and found none of them to be highly toxic to HeyA8 cells (*Figure 1—figure supplement 2B*). To determine their knockdown efficiency, we induced their expression in cells carrying sensor plasmids in which the fluorophore Venus was linked to either the CD95L or CD95 open reading frame (ORF). Expression of most of these miR-30-based shRNAs also did not efficiently silence Venus expression (*Figure 1—figure supplement 2C*). In contrast, two of our most toxic shRNAs shL3 and shR6 when expressed in the Tet-inducible pTIP vector not only killed HeyA8 cells, but also very efficiently suppressed Venus fluorescence in cells expressing the targeted Venus sensor (*Figure 1—figure supplement 2D*). These data suggest that the levels of shRNAs produced from the miR-30-based vector may not be sufficient to be toxic to the cancer cells. Because expression levels of shRNAs are difficult to titer, we used siRNAs to determine the concentration of the toxic CD95L-derived siL3 required to kill HeyA8 cells (*Figure 1—figure supplement 2E*). Growth was effectively blocked (and cells died, data not shown) when siL3 was transfected at 1 nM—a concentration well below the commonly used and recommended siRNA concentration of 5–50 nM)—but not at 0.1 nM. These data suggest that this form of toxicity does not require high amounts of si- or shRNAs; however, the low expression we achieved from the miR-30 based shRNA vectors was not enough to effectively induce the toxicity. Because these miR-30-based shRNA vectors were developed to reduce off-target effects, the toxicity of CD95 and CD95L-targeting si/shRNAs described by us and others could be due to an OTE. While this was a plausible explanation, the high percentage of toxic si/shRNAs derived from CD95 and CD95L seemed to exclude a standard OTE and pointed at a survival activity of CD95 and CD95L.

We therefore tested whether exogenously added recombinant CD95L protein could protect cells from the toxicity of CD95L-derived shRNAs. When NB7 cells were incubated with different concentrations of a soluble form of CD95L (S2), toxicity exerted by shL1 was not affected (*Figure 1A*, left panel). NB7 neuroblastoma cells were chosen for these experiments because they lack expression of caspase-8 (*Teitz et al., 2000*) and hence are completely resistant to the apoptosis-inducing effects of CD95L. An ostensible moderate and dose-dependent protection was detected when cells were treated with a highly active leucine-zipper tagged CD95L (LzCD95L) (*Figure 1A*, center panel). However, this effect is likely due to the growth-promoting activities of soluble CD95L, which also significantly affected the growth of the cells expressing a scrambled control shRNA (seen for both S2 and LzCD95L). The recombinant LzCD95L protein was active, as demonstrated by its apoptosis-inducing capacity in CD95 apoptosis-sensitive MCF-7 cells (*Figure 1A*, right panel).

To test whether CD95L or CD95 proteins could protect cancer cells from death, we introduced silent mutations into the targeted sites of three very toxic shRNAs: shL1, shL3 (both targeting CD95L) and shR6 (targeting CD95). We first introduced eight silent mutations into the sites targeted by either shL1 or shL3 (*Figure 1B*) and expressed these proteins in NB7 cells (*Figure 1C*). Both

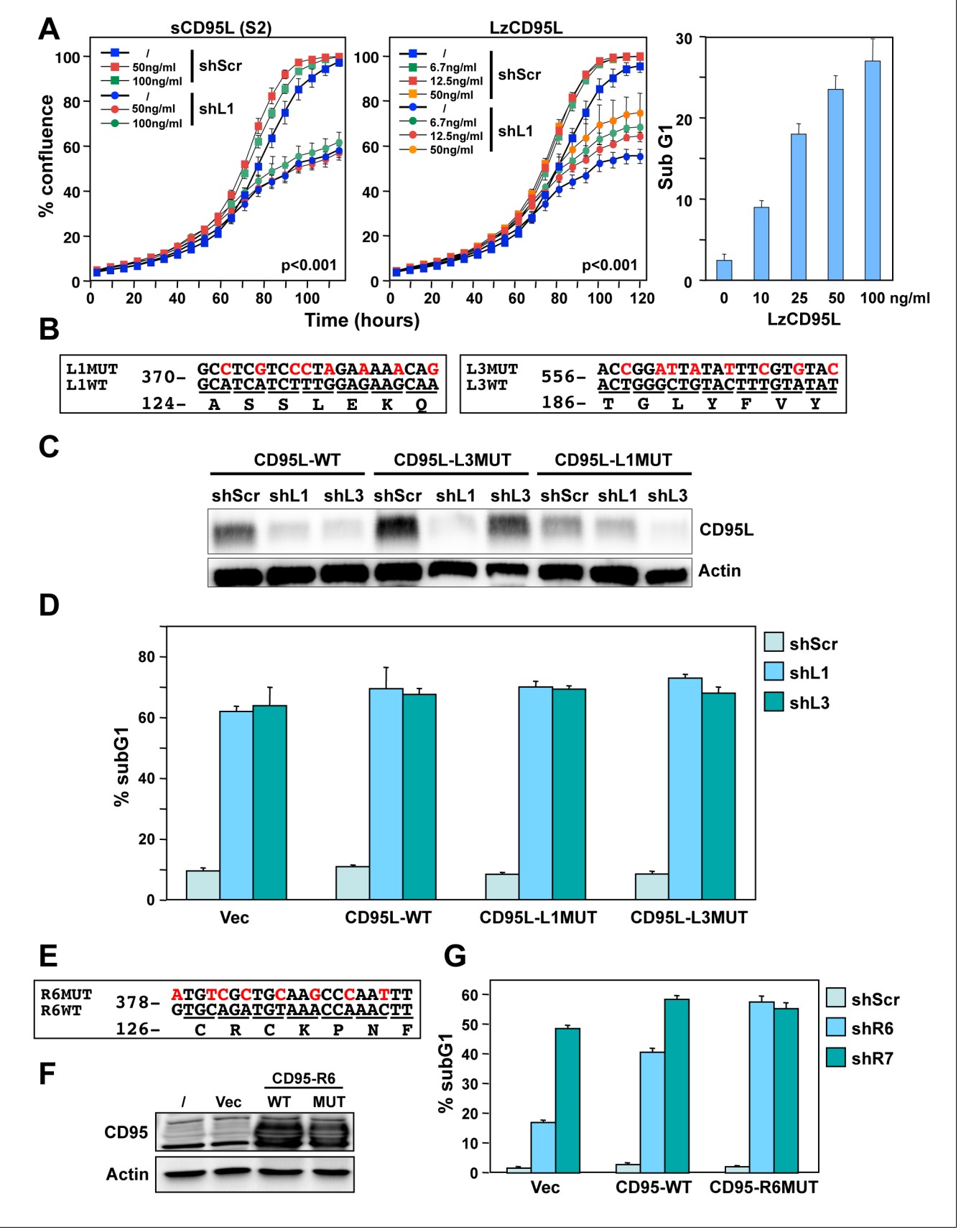

**Figure 1.** Exogenous CD95L or CD95 proteins do not protect cells from toxicity of CD95L/CD95-derived shRNAs. (**A**) *Left:* Percent cell confluence over time of NB7 cells after infection with either pLKO-shScr or pLKO-shL1 and concurrent treatment with different concentrations of soluble CD95L protein (S2). Two-way ANOVA was performed for pairwise comparisons of % confluence over time between shScr expressing cells untreated (/) or treated with 100 ng/ml S2. Each data point represents mean ±SE of three replicates. *Center:* Percent cell confluence over time of NB7 cells after infection with either pLKO-shScr or pLKO-shL1 and concurrent treatment with different concentrations of leucine zipper-tagged (Lz)CD95L protein. Two-way ANOVA was performed for pairwise comparisons of % confluence over time between shScr-expressing cells untreated or treated with 50 ng/ml LzCD95L. Each data point represents mean ±SE of three replicates. *Right:* Percent nuclear PI staining of MCF-7 cells 24 hrs after adding different amounts of LzCD95L. (**B**) Schematic of the eight silent mutations introduced to the shL1 and the shL3 target sites of CD95L. (**C**) Western blot analysis of CD95L and β-actin in NB7 cells over-expressing CD95L-WT, CD95L-L1MUT, or CD95L-L3MUT 3 days after infection with pLKO-shScr, pLKO-shL1, or pLKO-shL3. Shown is one of two repeats of this analysis. (**D**) Percent nuclear PI staining of NB7 cells expressing empty pLenti vector, CD95L-WT, CD95L-L1MUT, or CD95L-L3MUT 6 days after infection with either pLKO-shScr, pLKO-shL1, or pLKO-shL3. Each bar represents mean ±SD of three replicates. (**E**) Schematic of the eight silent mutations introduced at the shR6 site of CD95. (**F**) Western blot analysis of CD95 and β-actin in MCF-7 cells over-expressing CD95-WT or CD95-R6MUT. (**G**) Percent nuclear PI staining of MCF-7 cells expressing empty pLNCX2 vector, CD95-WT, or CD95-R6MUT 6 days after infection with pLKO-shScr, pLKO-shR6, or pLKO-shR7. Each bar represents mean ±SD of three replicates.

DOI: https://doi.org/10.7554/eLife.29702.003

The following figure supplements are available for figure 1:

**Figure supplement 1.** The majority of siRNAs and shRNAs targeting CD95L or CD95 are toxic.
DOI: https://doi.org/10.7554/eLife.29702.004

**Figure supplement 2.** Toxicity of si/shRNAs is dose dependent.
DOI: https://doi.org/10.7554/eLife.29702.005

mutant constructs were highly resistant to knockdown by their cognate shRNA but still sensitive to knockdown by the other targeting shRNA (*Figure 1C*). Overexpression of these shRNA-resistant versions of the CD95L ORF did not protect the cells from shL1 or shL3, respectively (*Figure 1D*). Interestingly, expression of full length CD95L slowed down the growth of the NB7 cells right after infection with the lentivirus despite the absence of caspase-8 (data not shown). Infection with shRNAs was therefore performed 9 days after introducing CD95L when the cells had recovered and expressed significant CD95L protein levels (*Figure 1C*). We then mutated the CD95 mRNA in the targeted site of shR6 (*Figure 1E*). Neither expression of wild-type (wt) nor mutated (MUT) CD95 in MCF-7 cells (*Figure 1F*) reduced the toxicity when cells were infected with the pLKO-shR6 or another toxic lentiviral shRNA, pLKO-shR7 (*Figure 1G*). These data suggested that neither exogenously added recombinant CD95L or exogenously expressed CD95L or CD95 protein can protect cells from toxic shRNAs derived from these genes.

To determine whether we could prevent cancer cells from dying by this form of cell death by deleting the endogenous targeted sites, we used CRISPR/Cas9 gene-editing to excise sites targeted by different shRNAs and siRNAs in both alleles of the CD95 and CD95L genes. We first deleted a 41 nt piece of the CD95L gene in 293T cells, that contained the target site for shL3 (*Figure 2A and C*). While internal primers could not detect CD95L mRNA in three tested clones, primers outside of the deleted area did detect CD95L mRNA (*Figure 2D*, and data not shown). Three clones with this shL3 Δ41 deletion were pooled and tested for toxicity by shL3 expressed from a Tet-inducible plasmid (pTIP-shL3). Compared to a pool of control cells transfected only with the Cas9 plasmid, the 293T shL3 Δ41 cells were equally sensitive to the toxic shRNA (*Figure 2G*). This was also observed when the clones were tested individually (data not shown).

To exclude the possibility that shL3 was inducing cell death due to a unique activity of shL3 and/ or 293T cells, we deleted the same 41 nt in CD95L in the ovarian cancer cell line HeyA8; We also generated HeyA8 clones in which we either removed a 64 nt region containing the target site for the siRNA siL3 in the CD95L coding sequence or a 227 nt region containing the target site for shR6 in CD95 (*Figure 2A and B* and *Figure 2—figure supplement 1*). In all cases, homozygous deletions were generated (*Figure 2E*). To confirm the deletion of the shR6 target site, we infected HeyA8 cells treated with the Cas9 plasmid only and HeyA8 with a homozygous deletion of the shR6 site with shR6 and, as positive controls, with shR2 (targeting the CD95 ORF) and shR6' (targeting the CD95 3'UTR). Five days after infection, CD95 mRNA was quantified by real time PCR using a primer located outside the 227 bp deletion (*Figure 2F*). The mutated CD95 mRNA was still detectable in the shR6 Δ227 cells. While shR2 and shR6' (both targeting outside the deleted region) caused knockdown of CD95 mRNA in both the Cas9 control and the shR6 Δ227 cells, shR6 could only reduce

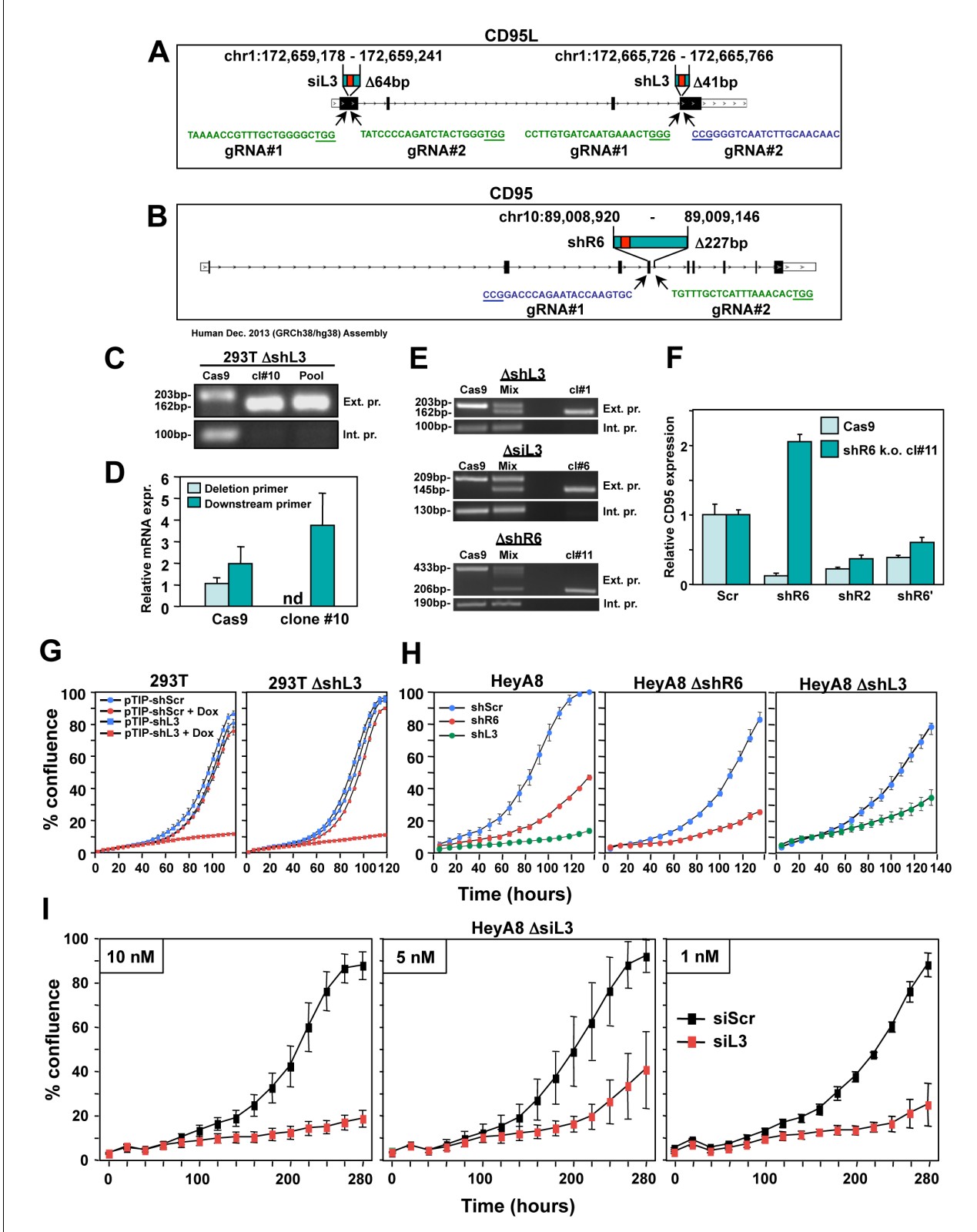

**Figure 2.** CD95 and CD95L derived si/shRNAs kill cells in the absence of the targeted sites in CD95 or CD95L. (**A**) Schematic of the genomic locations and sequences of the gRNAs used to excise the siL3 (Δ64bp) and shL3 (Δ41bp) target sites from CD95L. PAM site is underlined. Green indicates a gRNA targeting the sense strand. Blue indicates a gRNA targeting the antisense strand. (**B**) Schematic showing the genomic locations and sequences of the gRNAs used to excise the shR6 (Δ227bp) target site. Mix, pool of three 293T clones with the homozygous shL3 deletion. (**C**) PCR with flanking

*Figure 2 continued on next page*

*Figure 2 continued*

(*top panels*) and internal (*bottom panels*) primers used to confirm the Δ41 deletion in the shL3 site in one of the three homozygous deletion 293T clones generated. Cells transfected with Cas9 only (Cas9) are wild-type. (**D**) Quantitative PCR for endogenous CD95L with a primer downstream of the Δ41 shL3 deletion and another primer internal to the deleted region. nd, not detectable. Each bar represents mean ±SD of three replicates. (**E**) PCR with flanking (*top row*) and internal (*bottom row*) primers used to confirm the presence of the shL3 Δ41 (*top panel*), siL3 Δ64 (*middle panel*), and shR6 Δ227 (*bottom panel*) deletions in HeyA8 clones. Mix, HeyA8 cells after transfection with Cas9 and gRNAs but before single cell cloning. (**F**) Quantitative PCR for CD95 in HeyA8 cells transfected with Cas9 plasmid (Cas9) alone, or the HeyA8 ΔshR6 clone #11. RNA was extracted 5 days after infection with pLKO-shScr, pLKO-shR6, pLKO-shR2, or pLKO-shR6' (targeting the 3'UTR). Each bar represents mean ±SD of three replicates. (**G**) Percent cell confluence over time of 293T cells (*left*) and a pool of three 293T clones with a homozygous deletion of the shL3 target site (*right*) infected with pTIP-shScr or pTIP-shL3 and treatment with or without Dox. Data are representative of two independent experiments. Each data point represents mean ±SE of six replicates. (**H**) *Left:* Percent confluence over time of HeyA8 cells infected with pLKO-shScr, pLKO-shR6, or pLKO-shL3. *Center:* Percent confluence over time of a HeyA8 clone with a homozygous deletion of the shR6 target site infected with either pLKO-shScr or pLKO-shR6. *Right:* Percent confluence over time of a pool of three HeyA8 clones with a homozygous deletion of the shL3 site infected with either pLKO-shScr or pLKO-shL3. Data are representative of two independent experiments. Each data point represents mean ±SE of three replicates. (**I**) Percent confluence over time of a pool of three HeyA8 clones harboring a homozygous deletion of the siRNA siL3 target site after transfection with different concentrations of siScr or siL3. Data are representative of three independent experiments. Each data point represents mean ±SE of three replicates.

DOI: https://doi.org/10.7554/eLife.29702.006

The following figure supplement is available for figure 2:

**Figure supplement 1.** Knockout of CD95 in HeyA8 cells.

DOI: https://doi.org/10.7554/eLife.29702.007

mRNA expression in the Cas9 control cells. These data document that HeyA8 CD95 shR6 Δ227 cells no longer harbor the sequence targeted by shR6.

Now having HeyA8 cells lacking one of three RNAi-targeted sites in either CD95 or CD95L, we could test the role of the CD95 and CD95L gene products in protecting HeyA8 cells from the death induced by either shRNA (shL3 and shR6, two different vectors: pLKO or the Tet inducible pTIP) or the siRNA siL3. In all cases, the shRNA or siRNA that targeted the deleted region was still fully toxic to the target-site deleted cells (*Figure 2H and I*). We saw efficient growth reduction and cell death in siL3 site-deleted cells transfected with as little as 1 nM siL3 (*Figure 2I*, and data not shown). These data firmly establish that cells were not dying due to the knockdown of either CD95 or CD95L.

## Involvement of canonical RNAi

shRNAs and early generation naked siRNAs showed general toxicity when introduced in large amounts, presumably by eliciting an interferon (IFN) response (*Marques and Williams, 2005*) or by saturating the RISC (*Grimm et al., 2006*). However, both chemically modified siRNAs at very low concentrations and lentiviral shRNAs at an MOI <1 were still toxic (data not shown). We therefore decided to test whether the observed toxicity involved canonical RNAi and activity of the RISC. To test shRNAs or siRNAs targeting CD95L, we introduced the Venus-CD95L sensor (inset in *Figure 3A*, right panel) into HeyA8 CD95 protein k.o. cells we had generated in the process of deleting the shR6 site (*Figure 2—figure supplement 1*, clone #2 was used for the following studies; see figure legend for strategy and characterization of the clones). While double-stranded (ds)-siL3 effectively silenced Venus expression and induced toxicity, neither the sense nor the antisense single-stranded (ss)RNAs significantly decreased Venus expression or induced toxicity (*Figure 3A*). In addition, no activity was found when ds-siL3, synthesized as deoxyribo-oligonucleotides, was transfected into the cells (*Figure 3B*). Using this type of analysis, we tested a number of modified siRNAs for RNAi activity and toxicity. For siRNAs to be fully active, they require 3' overhangs on both strands (*Bernstein et al., 2001*). Converting siL3 to a blunt-end duplex resulted in substantial loss of RNAi activity and toxicity (*Figure 3C*). Due to the topology of the RISC, siRNA activity is decreased by modification of the 5' end of the antisense/guide strand (*Chiu and Rana, 2003*). To test whether cell death induced by siL3 would be affected by a bulky modification, we placed a Cy5 moiety at any of the four possible ends of the siL3 duplex. Only when the siL3 duplex carried a 5' modification in the guide strand did it prevent RNAi activity and toxicity; modifications in the three other positions had no effect (*Figure 3C*). This was confirmed for another siRNA, siL2. To test whether the toxicity of siL3 required association with a macromolecular complex, which would be consistent with RISC involvement, we performed a competition experiment. HeyA8 cells were transfected with 10 nM of

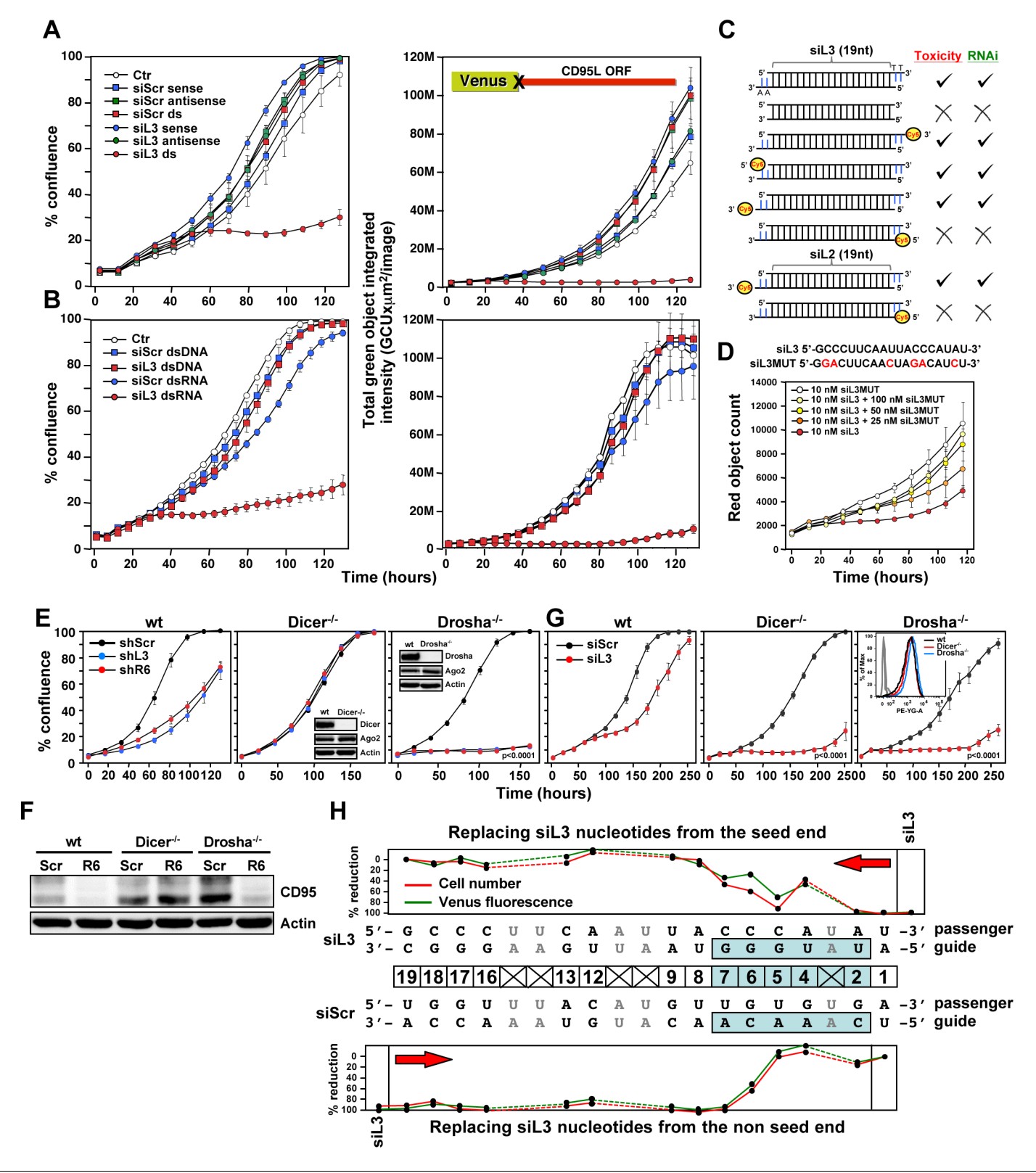

**Figure 3.** Toxicity of CD95L-derived siRNAs involves canonical RNAi activity. (**A**) Percent cell confluence (*left*) and total green object integrated intensity (*right*) over time of a HeyA8 CD95 knockout clone (ΔR6 cl#2) expressing the Venus-CD95L sensor either untreated (Ctr) or after transfection with 25 nM of single-stranded sense, single-stranded antisense, or double-stranded (ds) siScr or siL3 siRNAs. The CD95L sensor is schematically shown and comprises the Venus ORF fused to the CD95L ORF lacking the A of the ATG start codon (X). Data are representative of two independent experiments.

*Figure 3 continued on next page*

*Figure 3 continued*

Each data point represents mean ±SE of three replicates. (B) Percent cell confluence (*left*) and total green object integrated intensity (*right*) over time of the HeyA8 CD95L sensor cell used in *Figure 3A* after transfection with 5 nM siScr or siL3 double-stranded RNA (dsRNA) or double-stranded DNA (dsDNA). Data are representative of two independent experiments. Each data point represents mean ±SE of three replicates. (C) Summary of experiments to test whether siL3 and siL2 siRNAs modified as indicated (*left*) were active (check mark) or not (X) in reducing green fluorescence or cell growth (both >70% reduction at end point) when transfected at 25 nM (except for blunt end oligonucleotides which were used at 5 nM and compared to 5 nM of siL3) into HeyA8 CD95L sensor cells used in *Figure 3A*. Endpoints were 164 hrs for blunt end siRNA transfection, 180 hrs for modified siL3 and 144 hrs for modified siL2 siRNA transfections. Every data row is based on cell growth and green fluorescence quantification data executed as shown in **A**. Each analysis was done in triplicate and based on two independent repeats. (D) Red object count over time of HeyA8 cells (expressing NucRed) after transfection with different ratios of siL3 and mutant siL3 (siL3MUT). Data are representative of two independent experiments. Each data point represents mean ±SE of three replicates. (E) Percent cell confluence over time of HCT116 parental (*left*) or Dicer$^{-/-}$ (clone #43, another Dicer$^{-/-}$ clone, #45, gave a similar result, data not shown), or Drosha$^{-/-}$ (*right*) cells after infection with either shScr, shL3 or shR6 pLKO viruses. Inserts show the level of protein expression levels of Drosha/Dicer and AGO2 levels in the tested cells. Data are representative of three independent experiments. Each data point represents mean ±SE of four replicates. Drosha$^{-/-}$ cells were more sensitive to toxic shRNAs than wt cells (p<0.0001, according to a polynomial fitting model). (F) Western blot analysis of HCT116 wt, Dicer$^{-/-}$ or Drosha$^{-/-}$ cells 4 days after infection with either pLKO-shScr or pLKO-shR6. (G) Percent cell confluence over time of HCT116 wt, Dicer$^{-/-}$ (clone #43) and Drosha$^{-/-}$ cells after transfection with 25 nM siScr or siL3. Data are representative of four independent experiments (Dicer$^{-/-}$ clone #45, gave a similar result, data not shown). Each data point represents the mean ±SE of four replicates. Data in insert confirm similar uptake of transfected siRNA (25 nM of siGLO Red) into wild-type, Dicer$^{-/-}$ and Drosha$^{-/-}$ cells. Dicer$^{-/-}$ and Drosha$^{-/-}$ cells were more sensitive to siL3 than wt cells (p<0.0001, according to a polynomial fitting model). (H) Percent reduction in Venus expression (green) and in cell number (red object count [red]) over time of HeyA8 cells expressing the Venus-CD95L sensor and red nuclei after transfection with 5 nM of different chimeric siRNAs generated by substituting nucleotides in the toxic siL3 with the scrambled siRNA sequence beginning at either the seed match end (top) or the opposite end (bottom) of siL3 after 188 hr. The schematic in the middle shows the sequence of siL3 and the siScr siRNA (both sense and antisense strands). The 6mer seed sequence region of siL3 (positions 2 to 7) is highlighted in light blue. Nucleotides shared by siScr and siL3 are shown in grey font. Data are representative of two independent experiments. Each data point represents mean of three replicates. In another independent experiment cells were transfected with 25 nM with a very similar result (data not shown).

DOI: https://doi.org/10.7554/eLife.29702.008

siL3, and a mutated nontoxic oligonucleotide, siL3MUT, was titered in (*Figure 3D*). siL3MUT reduced the growth inhibitory activity of siL3 in a dose-dependent fashion suggesting that siL3 and siL3MUT compete for the same binding site in the cells, pointing at involvement of the RISC.

To determine involvement of RNAi pathway components in the toxicity of CD95 and CD95L-derived sequences, we tested HCT116 cells deficient for either Drosha or Dicer (*Kim et al., 2016*). Growth of parental HCT116 cells was impaired after infection with shL3 or shR6 viruses (*Figure 3E*, left panel). Consistent with the requirement of Dicer to process shRNAs, Dicer$^{-/-}$ cells were completely resistant to the toxic shRNAs (*Figure 3E*, center panel). This was also supported by the inability of shR6 to silence CD95 protein expression in these cells (*Figure 3F*). Dicer$^{-/-}$ cells were not resistant to toxic siRNAs as these cells died when transfected with siL3, which is consistent with mature siRNAs not needing further processing by Dicer (*Figure 3G*, center panel). Interestingly, Drosha$^{-/-}$ cells were hypersensitive to the two toxic shRNAs (*Figure 3E*, right panel, p<0.0001, according to a polynomial fitting model), and shR6 efficiently knocked down CD95 expression in Drosha$^{-/-}$ cells (*Figure 3F*). Both Drosha$^{-/-}$ and Dicer$^{-/-}$ cells were much more susceptible to the toxicity induced by siL3 than parental cells (*Figure 3G*, center and right panel, p<0.0001, according to a polynomial fitting model). The hypersensitivity of the Drosha$^{-/-}$ cells to toxic si/shRNAs and of Dicer$^{-/-}$ cells to toxic siRNAs can be explained by Drosha$^{-/-}$ and Dicer$^{-/-}$ cells allowing much more efficient uptake of mature toxic RNAi-active species into the RISC because they are almost completely devoid of competing endogenous miRNAs (*Kim et al., 2016*).

To determine the contribution of the siRNA seed sequence to their toxicity, we generated a set of chimeric siRNAs in which we systematically replaced nucleotides of the toxic siL3 siRNA with nucleotides of a nontoxic scrambled siRNA. We did this starting either from the seed end or from the opposite end (*Figure 3H*). HeyA8 cells expressing both the Venus-CD95L sensor (to monitor level of knockdown) and a Nuc-Red plasmid to fluorescently label nuclei (to monitor the effects on cell growth) were transfected with 5 nM of the chimeric siRNAs; total green fluorescence and the number of red fluorescent nuclei were quantified over time. The siL3 control transfected cells showed an almost complete suppression of the green fluorescence and high toxicity. In the top panel of *Figure 3H*, the data are summarized in which siL3 nucleotides were stepwise replaced with siScr nucleotides from the seed sequence end. Both RNAi and toxicity were profoundly reduced

when three of the terminal siL3 nucleotides were replaced with the siScr nucleotides in those positions, suggesting the seed region (6mer highlighted in blue) is critical for both activities. Consistently, as shown in the bottom panel of *Figure 3H*, when siL3 nucleotides were replaced with siScr nucleotides from the non-seed end, neither RNAi nor the toxicity was diminished until replacements affected residues in the seed region. These data suggest the 6mer seed sequence of siL3 was critical for both RNAi activity and its toxicity.

## Toxic si/shRNAs cause downregulation of survival genes

A general OTE by RNAi has been described (*Birmingham et al., 2006*; *Jackson et al., 2006*; *Lin et al., 2005*). However, this was been reported to cause toxicity in most cases, and the targeted mRNAs were difficult to predict (*Birmingham et al., 2006*). The fact that 22 of the tested CD95 and CD95L-targeting sh- and si/DsiRNAs were toxic to many cancer cells evoking similar morphological and biological responses (*Hadji et al., 2014*) generated a conundrum: Could an OTE trigger a specific biology? To test this, we expressed two toxic shRNAs - one targeting CD95L (shL3) and one targeting CD95 (shR6) - in cells lacking their respective target sequences and subjected the RNA isolated from these cells to an RNA-Seq analysis. In order to detect effects that were independent of cell type, delivery method of the shRNA, or targeted gene, we expressed shL3 in 293T (ΔshL3) cells using the Tet-inducible vector pTIP and shR6 in HeyA8 (ΔshR6) cells using the pLKO vector. In each case, changes in RNA abundance were compared to cells expressing a non-targeting shRNA in matching vectors. Total RNA was harvested in all cases at either the 50 hr time point (before the onset of cell death) or at the 100 hr time point (during cell death) (*Figure 4A*). To achieve high stringency, the data were then analyzed in two ways: first, using a conventional alignment-based analysis to identify genes for which the mRNA changed more than 1.5-fold (and an adjusted p-value of less than 0.05) and second, by a read-based method, in which we first identified all reads that changed >1.5 fold and then subjected each read to a BLAST search to identify the gene it was derived from. Only RNAs that were detected by both methods were considered (*Supplementary file 1*). The combination of the analyses resulted in one mRNA that was upregulated and 11 mRNAs that were downregulated (*Figure 4B*). Using an arrayed qPCR approach, most of these detected mRNA changes were validated for both cell lines (*Figure 4—figure supplement 1A*). Interestingly, for nine of the eleven genes, published data suggest they are either highly upregulated in cancer and/or critical for the survival of cancer cells, as their inhibition or knockdown resulted in either growth reduction or induction of various forms of cell death (see legend of *Figure 4—figure supplement 1* for details). Significantly, six of these eleven downregulated genes were recently identified in two independent genome-wide lethality screens to be critical for cancer cell survival (*Blomen et al., 2015*; *Wang et al., 2015*) (*Figure 4B* and *Figure 4—figure supplement 1B*) (*Supplementary file 2*). Considering these two screens only identified 6.6% of human genes to be critical for cell survival, we found a significant enrichment (54.5%, p-value=$3 \times 10^{-6}$ according to binomial distribution) of these survival genes among the genes downregulated during the cell death induced by either shL3 or shR6. All six survival genes are either highly amplified or mutated in human cancers (*Figure 4—figure supplement 2A*). In addition to these six genes, GNB1 and HIST1H1C were reported to be required fitness genes in a recent high-resolution CRISPR-based screen (*Hart et al., 2015*). A kinetic analysis showed most of the deregulated mRNAs were downregulated early with a significant effect already at 14 hr, more than two days before the onset of cell death (*Figure 4—figure supplement 1C* and data not shown). This suggested the cells were dying because of the silencing of multiple critical survival genes, providing an explanation for why multiple cell death pathways were activated. We therefore call this type of cell death DISE (for Death Induced by Survival gene Elimination).

To confirm some of the downregulated genes were also critical survival genes for HeyA8 cells, we transfected HeyA8 cells with siRNA SmartPools targeting each of the eleven genes. Individual knockdown of seven of the targeted genes resulted in reduced cell growth when compared to cells transfected with a pool of scrambled siRNAs (*Figure 4C*). To mimic the effect of the CD95 and CD95L-derived shRNAs, we treated HeyA8 cells with a combination of siRNA pools targeting these seven genes. Remarkably, 1 nM of this siRNA mixture (35.7 pM of each individual siRNA) was sufficient to effectively reduce growth of the cells (*Figure 4—figure supplement 2B*) and also cause substantial cell death (*Figure 4—figure supplement 2C*), suggesting it is possible to kill cancer cells with very small amounts of siRNAs targeting a network of these survival genes.

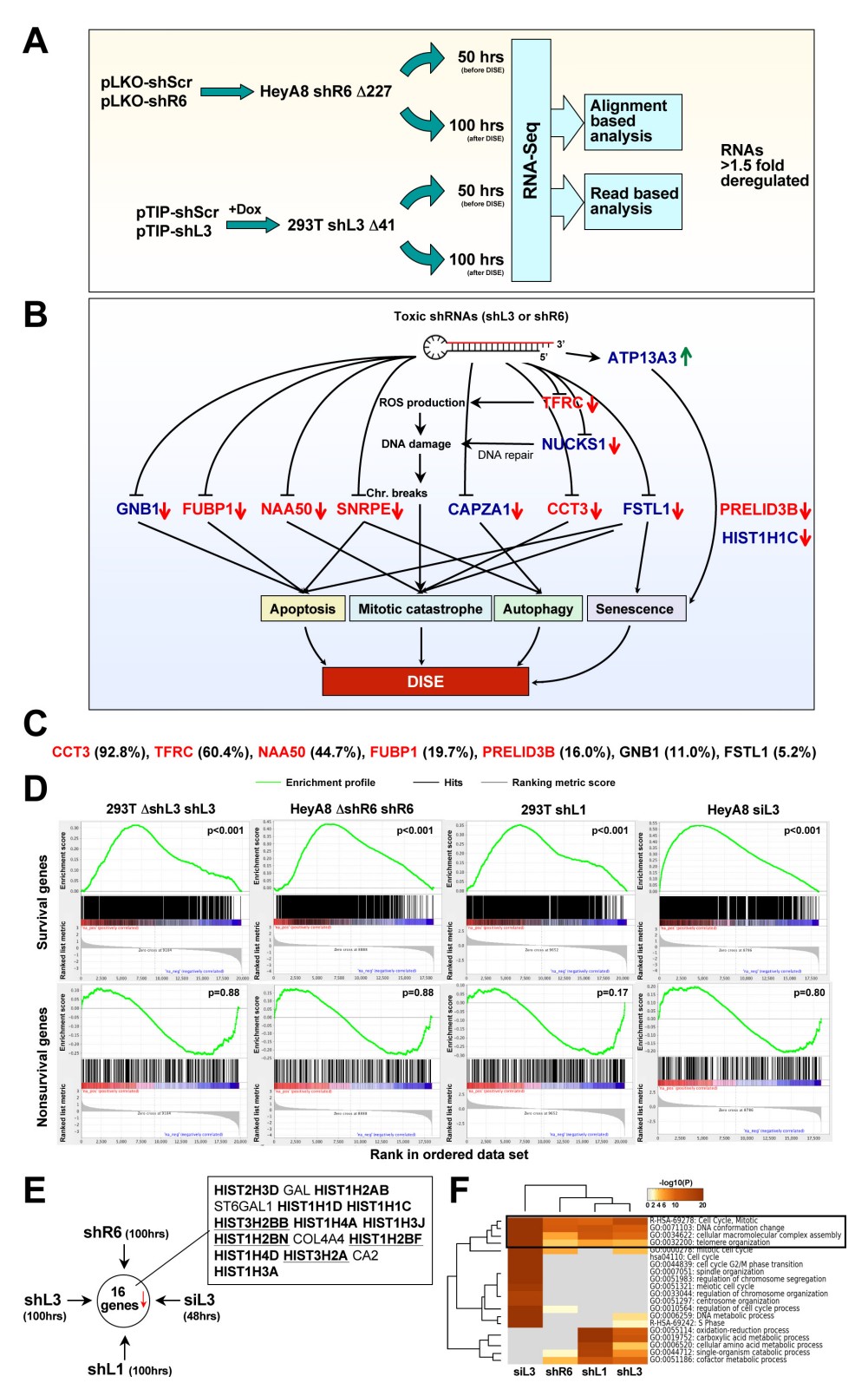

**Figure 4.** Toxic shRNAs derived from CD95 and CD95L cause downregulation of critical survival genes. (**A**) Schematic of RNA-Seq work flow for total RNA sample prepared both before (50 hr) and during (100 hr) DISE after expressing either shR6 or shL3 from different vector systems (i.e. pLKO-shR6 and pTIP-shL3) in different cells (HeyA8 shR6 Δ227 cells and 293T shL3 Δ41 cells). (**B**) One mRNA was up and 11 mRNAs were downregulated in the cells treated with toxic shL3 and shR6 as shown in *Figure 4A*. mRNAs shown in red were found to be essential cancer survival genes in two genome-wide

*Figure 4 continued*

lethality screens. The number of essential genes was enriched from 6.6% of the tested genes (*Blomen et al., 2015*; *Wang et al., 2015*) to 54.5% in our study (p=3×10⁻⁶ according to binomial distribution). (C) The level of growth inhibition observed in HeyA8 cells transfected with siRNA SmartPools (25 nM) individually targeting the listed survival genes. Targeting the seven genes shown significantly reduced cell growth compared to cells transfected with a siScr pool at 140 hrs (samples done in quadruplicate in two independent experiments) with an ANOVA p<0.05. (D) Gene set enrichment analysis for a group of 1846 survival genes (*top four panels*) and 416 nonsurvival genes (*bottom four panels*) identified in a genome-wide CRISPR lethality screen (*Wang et al., 2015*) after introducing Dox-inducible shL3 in 293T ΔshL3 cells (*left-most panels*), shR6 in HeyA8 ΔshR6 cells (*center-left panels*), shL1 in parental 293T cells (*center-right panels*), and siL3 in HeyA8 cells (*right-most panels*). Scrambled sequences served as controls. p-values indicate the significance of enrichment. (E) Schematics showing all RNAs at least 1.5 fold downregulated (adj p-value<0.05) in cells treated as in *Figure 4A*. Histones that are underlined contain a 3'UTR. (F) Metascape analysis of the 4 RNA Seq data sets analyzed. The boxed GO term clusters were highly enriched in all data sets.

DOI: https://doi.org/10.7554/eLife.29702.009

The following figure supplements are available for figure 4:

**Figure supplement 1.** Down-regulation of critical survival genes after treatment with CD95 and CD95L-derived shRNAs and siRNAs.

DOI: https://doi.org/10.7554/eLife.29702.010

**Figure supplement 2.** Characterization of the six genes downregulated in shL3 and shR6-treated cells and found to be critical survival genes in lethality screens.

DOI: https://doi.org/10.7554/eLife.29702.011

**Figure supplement 3.** Histones are downregulated in all forms of DISE but are not the most highly expressed genes in cells.

DOI: https://doi.org/10.7554/eLife.29702.012

To test the generality of this phenomenon, we inducibly expressed another CD95L derived shRNA, shL1, in 293T cells using the pTIP vector, and transfected HeyA8 cells with 25 nM siL3. We subjected the cells to RNA-Seq analysis 100 hrs and 48 hrs after addition of Dox or after transfection, respectively. To determine whether survival genes were downregulated in all cases of sh/siRNA-induced cell death, we used a list of 1882 survival genes and 423 genes not required for survival (nonsurvival genes) recently identified in a CRISPR lethality screen (*Supplementary file 2*). We subjected the four ranked RNA-Seq data sets to a gene set enrichment analysis using the two gene sets (*Figure 4D*). In all cases, survival genes were significantly enriched towards the top of the ranked lists (most downregulated). In contrast, nonsurvival genes were not enriched. One interesting feature of DISE that emerged was the substantial loss of histones. Of the 16 genes that were significantly downregulated in cells treated with any of the four sh/siRNAs, 12 were histones (*Figure 4E*). While it might be expected that dying cells would downregulate highly expressed genes such as histones, we believe that losing histones is a specific aspect of DISE because a detailed analysis revealed the downregulated histones were not the most highly expressed genes in these cells (*Figure 4—figure supplement 3*). In addition, almost as many genes with similarly high expression were found to be upregulated in cells after DISE induction.

A Metascape analysis revealed genes involved in mitotic cell cycle, DNA conformation change, and macromolecular complex assembly were among the most significantly downregulated across all cells in which DISE was induced by any of the four sh/siRNAs (*Figure 4F*). These GO clusters are consistent with DISE being a form of mitotic catastrophe with cells unable to survive cell division (*Hadji et al., 2014*) and suggest a general degradation of macromolecular complexes.

## Toxic si/shRNAs target survival genes in their 3'UTR

To test whether the toxic shRNAs directly targeted genes through canonical RNAi, we subjected the two gene lists obtained from the RNA-Seq analysis (the cell lines treated with either shL3 or shR6 at the 50 hr time point) to a Sylamer analysis (*van Dongen et al., 2008*) designed to find an enrichment of miRNA/siRNA-targeted sites in the 3'UTR of a list of genes ranked according to fold downregulation (*Figure 5A*). This analysis identified a strong enrichment of the cognate seed match for shL3 and shR6 in cells treated with either of these two shRNAs. The analyses with cells treated with shRNAs for 100 hrs looked similar but less significant, suggesting early targeting by the shRNAs followed by secondary events (data not shown). Enrichment in 6mers and 8mers were both detected (only 8mers shown) in the 3'UTRs but not the ORF of the ranked genes (data not shown).

Interestingly, the seed matches detected by the Sylamer analysis were shifted by one nucleotide from the expected seed match based on the 21mer coded by the lentivirus. RNA-Seq analysis

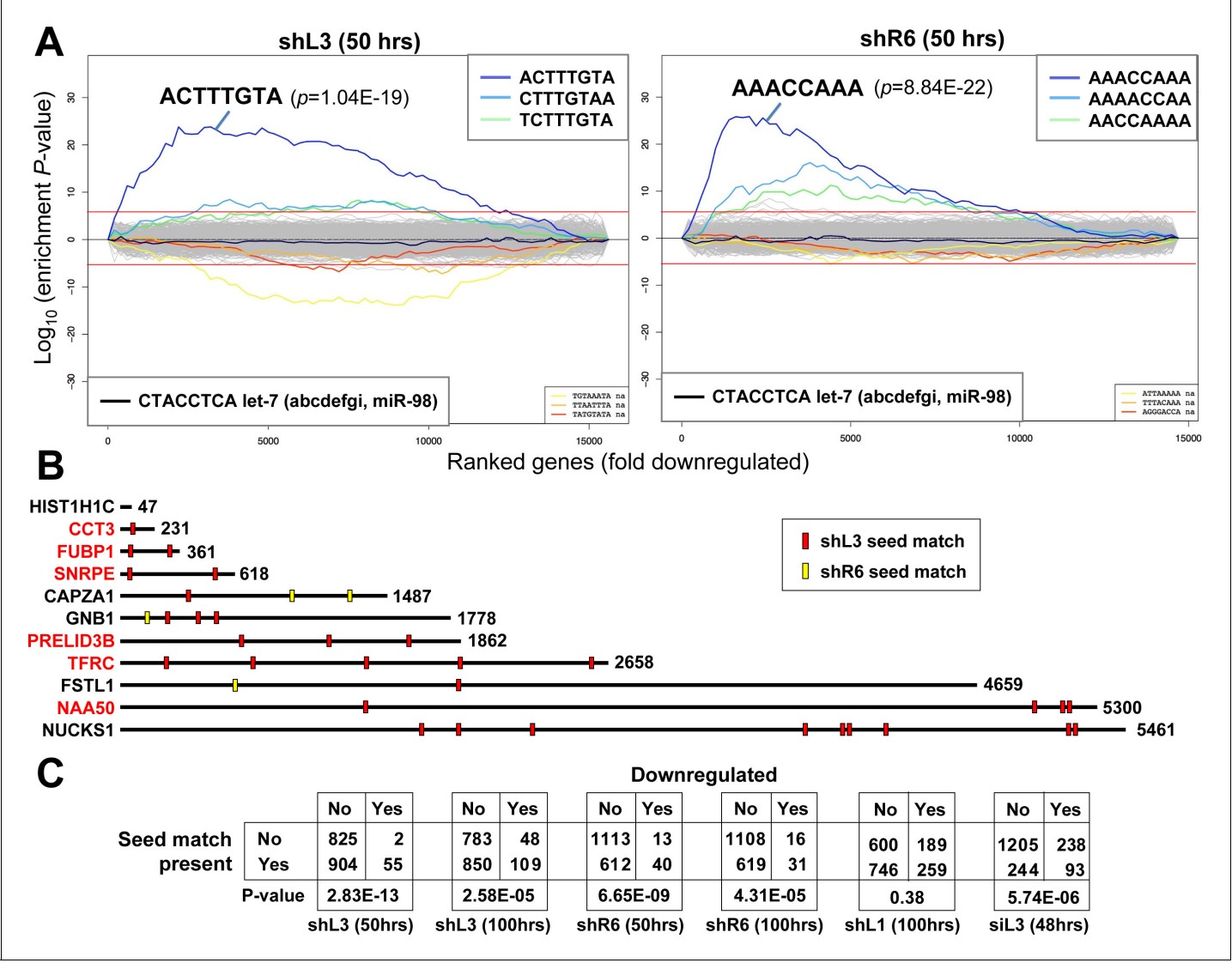

**Figure 5.** DISE inducing si/shRNAs target critical survival genes through RNAi. (**A**) Sylamer plots for the list of genes in the shL3 experiment (left) and the shR6 experiment (right) ordered from down-regulated to up-regulated. The most highly enriched sequence is shown which in each case is the 8mer seed match of the introduced shRNA. The red line corresponds to a p-value threshold of 0.05 after Bonferroni correction for the number of words tested (65536). Bonferroni-adjusted p-values are shown. The unadjusted p-values are 1.58E-24 and 1.35E-26, respectively. The black line represents the sequences carrying the let-7 8mer seed match. (**B**) Location of the 6mer seed matches of either shL3 or shR6 in the 3'UTRs of the 11 genes (shown at scale) identified in the RNA-Seq experiment described in *Figure 4A*. Red font indicates a critical survival gene. (**C**) A series of six 2 × 2 contingency tables comparing whether or not a critical survival gene is downregulated after treatment with the indicated siRNA or shRNA to whether or not its 3'UTR contains at least one seed match for the introduced sh/siRNA. p-values were calculated using Fisher's Exact Test to determine any significant relationship between gene downregulation and presence of seed matches in 3'UTR.

DOI: https://doi.org/10.7554/eLife.29702.013

The following figure supplements are available for figure 5:

**Figure supplement 1.** Quantification of the mature shRNA forms.
DOI: https://doi.org/10.7554/eLife.29702.014

**Figure supplement 2.** Identification of seed matches targeted by shL1 and siL3.
DOI: https://doi.org/10.7554/eLife.29702.015

**Figure supplement 3.** Activity to knockdown CD95 does not determine shRNA toxicity.
DOI: https://doi.org/10.7554/eLife.29702.016

performed for the small RNA fraction confirmed in all cases (shScr and shL3 in pTIP, and shScr and shR6 in pLKO), the shRNAs in the cells were cleaved in a way resulting in the predominant formation of an siRNA shifted one nucleotide away from the shRNA loop region (black arrow heads in *Figure 5—figure supplement 1A*). This allowed us to design toxic mature siRNAs based on the sequences of shL3 and shR6. These shRNA-to-siRNA converts were toxic to HeyA8 cells (*Figure 5—figure supplement 1B*) confirming that the observed toxicity was not limited to the TRC shRNA platform, but based on a sequence-specific activity of the si/shRNAs.

The generalizability of the Sylamer results for shL3 and shR6 was tested with cells treated with either shL1 or siL3. In both cases, when the ranked RNA Seq data were subjected to a Sylamer analysis, the seed matches of the si/shRNA introduced were again significantly enriched in the 3'UTR of downregulated RNAs (*Figure 5—figure supplement 2*). In none of the Sylamer analyses of the four data sets, did we see enrichment of seed matches in the 3'UTRs of downregulated RNAs that matched the passenger strand. In all cases, the only significantly enriched sequences matched the seed sequences in the guide strand of the si/shRNAs we introduced.

Our data suggested that DISE inducing si/shRNAs caused an early loss of survival genes, and at the same time downregulated RNAs through canonical RNAi targeting their 3'UTR. However, it was not clear whether the most highly downregulated survival genes were targeted in their 3'UTR by RNAi-active sequences. We determined as little as six nucleotides dictated whether an siRNA killed cancer cells (see *Figure 3H*). 10 of the 11 downregulated genes identified in the RNA-Seq analysis described in *Figure 4A and B* contained multiple 6mer seed matches for either shL3 and/or shR6 (*Figure 5B*). It is therefore likely the two shRNAs, shL3 and shR6, killed cells by targeting a network of genes enriched in critical survival genes through RNAi. The only gene without an shL3 or shR6 seed match was HIST1H1C. Interestingly, only four of the histones downregulated in cells after treatment with any of the four tested si/shRNAs had a 3'UTR (underlined in *Figure 4E*) suggesting that most histones were not directly targeted by the si/shRNAs.

Using arrayed qPCR, we tested whether other toxic shRNAs targeting either CD95 or CD95L also caused downregulation of some of the 11 genes silenced by shL3 and shR6. HeyA8 cells were transfected with the toxic siRNA siL3 (RNA harvested at 80 hrs) or the toxic shRNAs shL1, shL3 or shR7 (RNA harvested at 100 hrs). While shL1 did not have much of an effect on the expression of these genes, shR7 caused downregulation of 7 of 11 of the same genes targeted by shL3 even though the 6mer seed matches of the two shRNAs are very different (CTTTGT for shL3 and GGAGGA for shR7) (*Figure 4—figure supplement 1D*).

To determine whether preferential targeting of survival genes was responsible for the death of the cells, we tested whether there was an association between the presence or absence of a predicted seed match in the 3'UTR for the si/shRNA introduced and whether a gene would be downregulated (>1.5 fold downregulated, $p < 0.05$) among survival genes using the Fisher's Exact test (*Figure 5C*). In almost all cases, this analysis revealed that survival genes containing a predicted seed match in their 3'UTR were statistically more likely to be downregulated than survival genes without such a motif. The analysis with shL1 treated cells did not reach statistical significance, likely due to the fact that this shRNA was found to be very toxic and the 100 hr time point may have been too late to observe evidence of significant targeting. This interpretation is supported by the observation that the significance for both shL3 and shR6 to target survival genes was higher at 50 hrs when compared to the 100 hr time points (*Figure 5C*) and that the Sylamer analysis of the shL1 treated cells was less significant after 100 hrs of treatment than any of the other Sylamer analyses (*Figure 5—figure supplement 2*).

Now that we had established that the toxicity of the studied shRNAs involved targeting of survival genes rather than CD95 or CD95L, we had to assume that when studying a larger set of shRNAs that the level of knockdown of the targeted genes and the toxicity were not strictly correlated. This was confirmed for the TRC shRNAs targeting the ORF or 3'UTR of CD95 in CD95 high expressing HeyA8 cells (*Figure 5—figure supplement 3*). While some of the toxic shRNAs efficiently silenced CD95 (i.e. shR6 and shR2) few did not (i.e. shR5). In summary, our analyses suggest that cells die by DISE due to an early and selective silencing of survival genes through targeting seed matches in their 3'UTR followed by the downregulation of histones.

## Identification of toxic shRNAs in the CD95L and CD95 mRNAs

The majority of commercially available si-, Dsi-, and shRNAs targeting either CD95 or CD95L were highly toxic to cancer cells. We therefore asked whether these two genes contained additional sequences with similar activity. To test all shRNAs derived from either CD95L or CD95, we synthesized all possible shRNAs, 21 nucleotides long, present in the ORF or the 3'UTR of either CD95L or CD95 starting with the first 21 nucleotides after the start codon, and then shifting the sequence by one nucleotide along the entire ORF and 3'UTR (*Figure 6A*). We also included shRNAs from a gene not expressed in mammalian cells and not expected to contain toxic sequences, Venus. All 4666 oligonucleotides (700 Venus, 825 CD95L ORF, 837 CD95L 3'UTR, 987 CD95 ORF, and 1317 CD95 3'UTR shRNAs) were cloned into the Tet-inducible pTIP vector (*Figure 6B*) as five individual pools. We first tested the activity of each individual pool to be toxic and to target the Venus sensor protein (fused to either the ORF of CD95 or CD95L). NB7 cells were again used because of their resistance to the Venus-CD95L sensor, which was found to be slightly toxic to CD95 apoptosis competent cells. NB7-Venus-CD95L cells infected with the Venus-targeting shRNA pool showed some reduction in fluorescence when Dox was added, however, the shRNA pool derived from the CD95L ORF was much more active in knocking down Venus (*Figure 6—figure supplement 1A*). No significant green fluorescence reduction was detected in cells after infection with the shRNA pool derived from the CD95L 3'UTR since the targeted sequences were not part of the sensor. Similar results were obtained when NB7-Venus-CD95 cells were infected with the Venus, CD95 ORF, and CD95 3'UTR targeting shRNA pools. To determine their ability to reduce cell growth (as a surrogate marker for toxicity), we infected NB7 parental cells with each of the five pools (parental cells were used for this experiment to avoid a possible sponge effect by expressing either CD95L or CD95 sequences that were part of the Venus sensors). Interestingly, the pool of 700 shRNAs derived from Venus did not cause any toxicity (*Figure 6—figure supplement 1B*). In contrast, the pool of the shRNAs derived from CD95L significantly slowed down growth, while no toxicity was observed when cells were infected with the pool of shRNAs derived from the CD95L 3'UTR. In the case of CD95, both the shRNAs derived from the ORF and the 3'UTR showed some toxicity. However, the shRNAs derived from the 3'UTR caused greater toxicity compared to those derived from the ORF. The data suggest that overall the shRNAs derived from the CD95L ORF and the CD95 3'UTR contain the most toxic sequences.

To determine the toxicity of each of the shRNAs in the pools, NB7 cells were infected with the libraries of shRNA viruses (MOI <1), and after puromycin selection cells were pooled 1:1:1 (Venus ORF/CD95L ORF/CD95L 3'UTR pools or Venus ORF/CD95 ORF/CD95 3'UTR pools) to allow for competition between shRNAs when Dox was added (*Figure 6B*). Cells were cultured for 9 days with and without Dox to allow for cell death to occur. To identify depleted shRNAs, shRNA barcodes were detected through next generation sequencing of PCR products to determine the relative abundance of each shRNA in three pools: 1) the cloned plasmid libraries, 2) cells after infection and culture for 9 days without Dox, and 3) cells infected and cultured with Dox for 9 days. A total of 71,168,032 reads were detected containing a complete sequence of one of the cloned shRNAs. Virtually all shRNAs were substantially represented in the cloned plasmids (*Supplementary file 3*). The shRNAs in the CD95L pool (comprised of the Venus, CD95L ORF, and CD95L 3'UTR subpools) and the CD95 pool (comprised of the Venus, CD95 ORF, and CD95 3'UTR subpools) were ranked from highest (most toxic) to lowest underrepresentation. During this and subsequent analyses, we noticed in many cases, Dox addition did cause a reduction of shRNAs, indicating an increase in toxicity; however, in other instances, infection alone and without the addition of Dox was toxic. This effect was likely due to the well-described leakiness of the Tet-on system (*Pham et al., 2008*), which we confirmed for shR6 in NB7 cells (*Figure 6—figure supplement 2A*). To capture all toxic shRNAs, we therefore decided to split the analysis into two halves: 1) the changes in abundance after infection compared to the composition in the plasmid pool (infection -Dox) and 2) the changes in abundance after Dox addition compared to the infected –Dox cells (infection +Dox). In subsequent analyses, shRNAs underrepresented after infection are either boxed (*Figure 6C*) or shown (*Figures 6D and 7B and Figure 7—figure supplement 1B*) in blue and the ones underrepresented after Dox addition are either boxed or shown in orange. The results for all shRNAs are shown in *Figure 6—figure supplement 2B*. Grey dots represent all shRNAs and red dots represent only the ones that were significantly underrepresented at least 5-fold. Interestingly, the highest abundance of downregulated

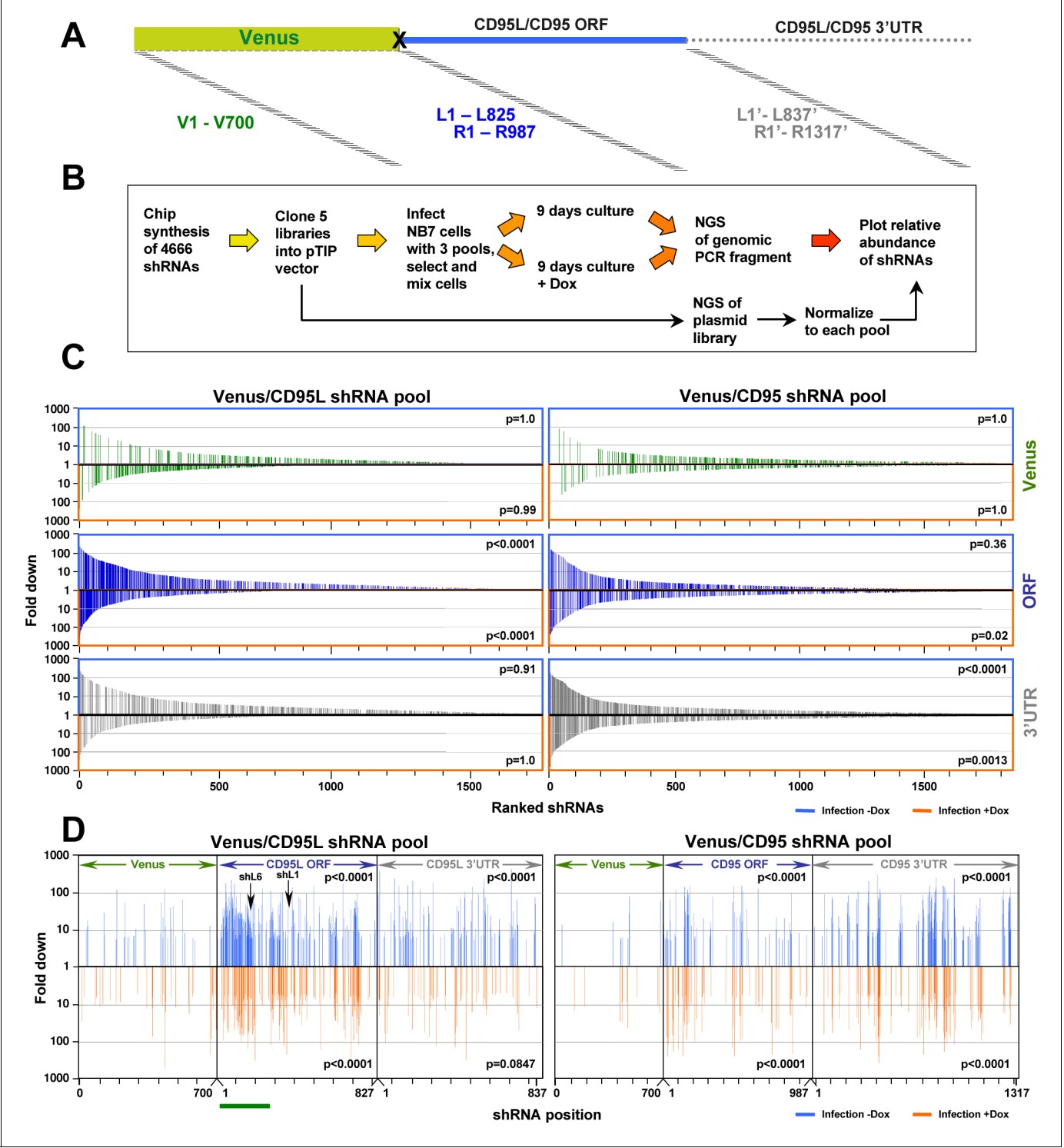

**Figure 6.** Identifying all toxic shRNAs derived from CD95L and CD95. (**A**) Schematic showing the cloned shRNAs covering the ORF of Venus and the ORFs and 3'UTRs of CD95L and CD95. The 3'UTR is displayed as a dashed line because it was not included in the full-length Venus-CD95L/CD95 sensors. (**B**) Work-flow of pTIP-shRNA library synthesis, shRNA screen and data analysis. (**C**) Ranked fold reduction of shRNAs spanning Venus and CD95L (ORF and 3'UTR) (*left three panels*) and Venus and CD95 (ORF and 3'UTR) (*right three panels*). The ranked lists were separated into the shRNAs derived from Venus (top), the ORFs (center) and the 3'UTRs (bottom). The p-value of enrichment for each ranked set of shRNAs is given. Only the parts of the ranked lists are shown with the downregulated shRNAs. For all six panels, the top section of each panel (boxed in blue) contains the data on shRNAs downregulated after infection of cells and cultured for 9 days without Dox when compared to the composition of the shRNA plasmid library and the bottom half (boxed in orange) contains the data on shRNAs downregulated after culture with Dox for 9 days when compared to the culture

*Figure 6 continued on next page*

*Figure 6 continued*

without Dox. P-values were calculated using Mann Whitney U tests with a one-sided alternative that the rank was lower. (D) The location of all shRNAs significantly downregulated at least five fold along the sequences of Venus, CD95L ORF, CD95L 3'UTR (left panel) and Venus, CD95 ORF, and CD95 3'UTR (right panel). The top half of each sub panel (blue ticks) shows the shRNAs downregulated after infection and the bottom half (orange ticks) contains the data on shRNAs downregulated after culture with Dox for 9 days. Significance of enrichment in the different subpanels is shown. p-values were calculated according to statistical tests of two proportions. Each data set was compared to the corresponding Venus distribution. Green line: sequence that corresponds to the intracellular domain of CD95L.

DOI: https://doi.org/10.7554/eLife.29702.017

The following figure supplements are available for figure 6:

**Figure supplement 1.** Toxicity and RNAi of individual shRNA pools.

DOI: https://doi.org/10.7554/eLife.29702.018

**Figure supplement 2.** Fold change in shRNA representation after infection of NB7 cells and after treatment with Dox.

DOI: https://doi.org/10.7554/eLife.29702.019

shRNAs was found in the CD95L ORF and the CD95 3'UTR pools of shRNAs, which is consistent with the increased toxicity observed when NB7 cells were infected with either of these two pools individually (see *Figure 6—figure supplement 1B*). The shRNAs of these two toxic pools were highly enriched in the underrepresented shRNAs in the two pooled experiments (CD95L and CD95). Their toxicity was also evident when all shRNAs in each pool (2362 shRNAs in the CD95L and 3004 shRNAs in the CD95 pool) were ranked according to the highest fold downregulation (*Figure 6C*). The three subpools in each experiment are shown separately. Thus, again this analysis identified the ORF of CD95L and the 3'UTR of CD95 as the subpool in each analysis with the highest enrichment of underrepresented shRNAs (*Figure 6C*).

This analysis allowed us to describe the toxicity landscape of CD95L and CD95 ORFs and their 3'UTRs (*Figure 6D*). All shRNAs significantly underrepresented at least five-fold (red dots in *Figure 6—figure supplement 2B*) are shown along the CD95L pool (*Figure 6D*, left) and the CD95 pool (*Figure 6D*, right) sequences. For both CD95L and CD95, toxic shRNAs localized into distinct clusters. The highest density of toxic sequences was found in the stretch of RNA that codes for the intracellular domain of CD95L (underlined in green in *Figure 6D*).

## Predicting shRNA toxicity - the toxicity index (TI) and GC content

Our data suggest toxic shRNAs derived from either CD95L or CD95 kill cancer cells by targeting a network of genes critical for survival through canonical RNAi. Therefore, we wondered how many 8mer seed sequences derived from these toxic shRNAs would have corresponding seed matches in the 3'UTR of critical survival genes in the human genome. Would it be possible to predict with some certainty in an *in silico* analysis what shRNAs would be toxic to cells? To calculate such a hypothetical toxicity index, we used the ranked CRISPR data set (*Wang et al., 2015*) with 1882 survival genes (SGs) and 423 nonSGs. Based on our RNA-Seq analyses, we hypothesized the survival genes contained more putative seed matches for toxic shRNAs in their 3'UTRs than the nonsurvival genes (*Figure 7A*, left) and that the number of seed matches in the 3'UTRs of survival genes divided by the number of seed matches in the 3'UTR of nonsurvival genes would, to some extent, predict toxicity of an si/shRNA (*Figure 7A*, right).

To establish a Toxicity Index (TI) for each shRNA, we first gathered 3'UTR sequences for 1846 of the survival genes and 416 of the nonsurvival genes. We then generated a list containing a normalized ratio of occurrences of every possible 8mer seed match in the 3'UTRs of the survival and nonsurvival gene groups. This resulted in a ratio for each of the 65,536 possible 8mer combinations (*Supplementary file 4*), the TI. We then assigned to each of the 4666 shRNAs in our screen its TI, and ranked each pool within the two experiments of our screen according to the highest TI (red stippled lines in *Figure 7B*). We then further separated the shRNAs into two groups: those that were toxic just after infection and those toxic after addition of Dox (*Figure 7B*, *Supplementary file 5*). In each ranked list, we could now assess whether the experimentally determined toxicity of shRNAs correlated with the *in silico* predicted TI. Remarkably, the highest enrichment of toxic shRNAs was found amongst those with higher TI for the subpool of shRNAs targeting the CD95L ORF followed by shRNAs in the subpool targeting the CD95 3'UTR. To confirm the significance of this finding, we repeated the analysis 10,000 times by randomly assigning 8mers and their associated TIs to the two

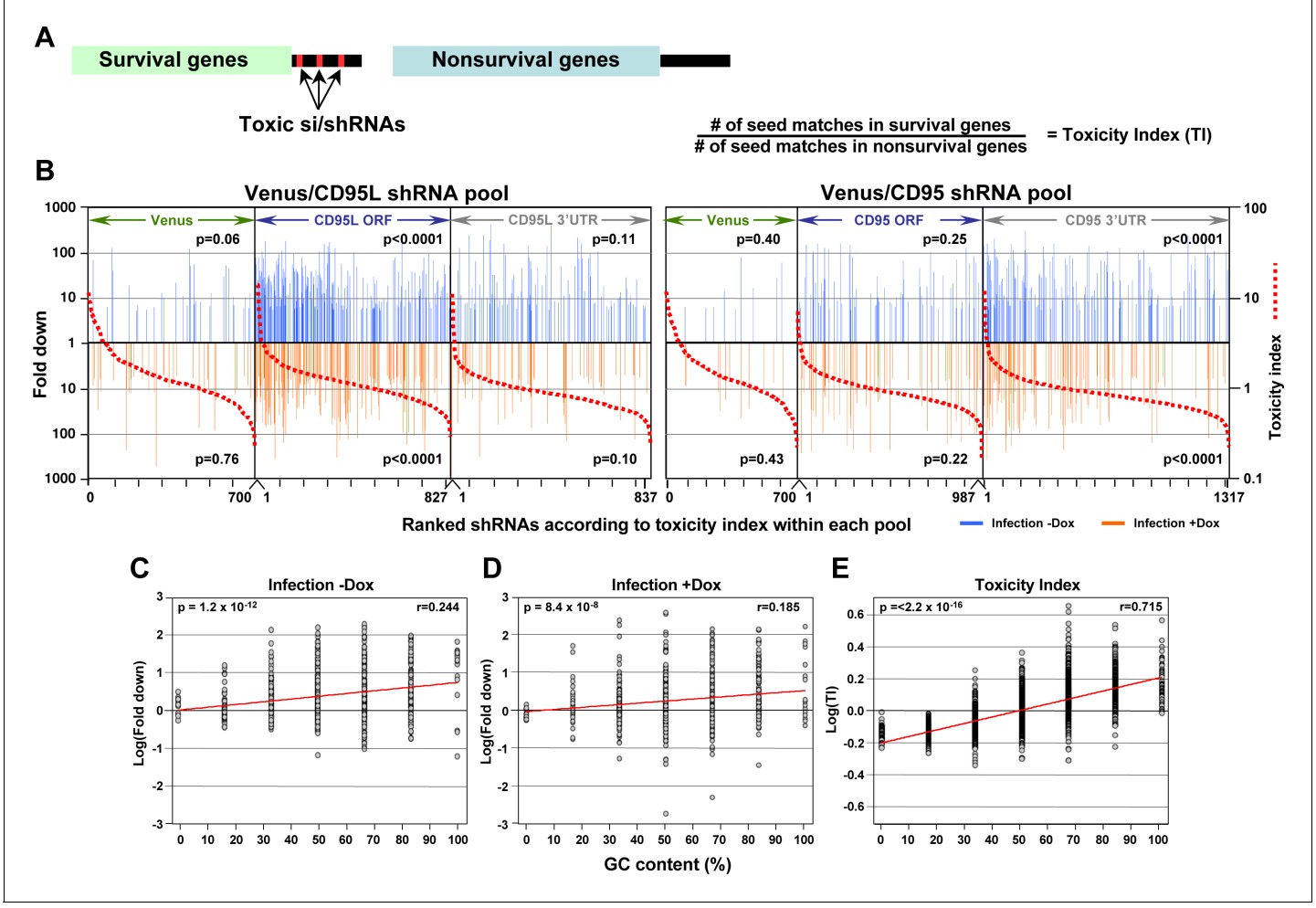

**Figure 7.** In silico prediction of DISE activity tracks with experimental determined toxicity of shRNAs. (A) *Left:* Schematic showing the preferential targeting of seed matches present in the 3′UTRs (red marks) of survival genes by toxic si/shRNAs. *Right:* The toxicity index (TI) is the normalized ratio of the number of 6mer or 8mer seed matches present in a list of survival genes versus a list of nonsurvival genes. (B) Fold downregulation versus ranked (8mer seed matched based) Toxicity Index for shRNAs of the Venus/CD95L pool (*left three panels*) and the Venus/CD95 pool (*right three panels*). Orange and blue tick marks indicate the same as in *Figure 6D*. To test if higher TI is enriched in shRNAs that were highly downregulated, p-values were calculated based on permutated datasets using Mann-Whitney U tests. The ranking of TI was randomly shuffled 10,000 times and the W statistic from our dataset was compared to the distribution of the W statistic of the permutated datasets. (C, D) Plot of fold downregulation of toxic shRNAs derived from CD95L ORF of the toxicity screens -Dox (left) or +Dox (center) versus GC content the 6mer seed in each shRNA. (E) Plot of the log(TI) of all 4092 possible 6mers versus GC content of the seeds. Pearson correlation coefficient and significance (p values) are given.

DOI: https://doi.org/10.7554/eLife.29702.020

The following figure supplement is available for figure 7:

**Figure supplement 1.** DISE does not just target all highly expressed genes.

DOI: https://doi.org/10.7554/eLife.29702.021

shRNA pools and again sorted the data from highest to lowest TI. The reported p-values were calculated based on these permutated datasets using Mann-Whitney U tests.

We noticed that survival genes tend to be more highly expressed than nonsurvival genes (data not shown). To address the question whether toxic si/shRNAs only target survival genes or all genes that are highly expressed, we recalculated the TI based on a set of 850 highly expressed and expression matched survival and nonsurvival genes (*Figure 7—figure supplement 1A*). This alternative TI tracked slightly less well with the toxic shRNAs we identified, but the enrichment of toxic shRNAs towards the top of the list ranked according to the new TI was still statistically significant (*Figure 7— figure supplement 1B*). This analysis demonstrates survival genes contain more seed matches for

toxic shRNAs in their 3'UTR than nonsurvival genes regardless of the expression level. This suggests, to a certain extent, it is possible to predict the experimental toxicity of shRNAs based on the *in silico* calculated TI.

Our data suggest DISE results from a sequence-specific off-target activity that depends on the presence of certain seed matches in the 3'UTR of survival genes. Thus, DISE inducing RISC associated small RNAs behave in manner similar to miRNAs. This raised the question whether these seed matches have special properties. While we did not find a sequence motif that was present in all toxic si/shRNAs, we did find that sequence composition, specifically GC content, which has been reported to affect the specificity of shRNAs (*Gu et al., 2014*; *Ui-Tei et al., 2004*), correlated with the toxicity of shRNAs. When the GC content of the 6mer seed sequences of all underrepresented shRNAs detected in the shRNA screen across the CD95L ORF was plotted, we found a significant correlation between the GC content and higher toxicity (indicated by underrepresentation) (*Figure 7C and D*). This correlation was even more pronounced when plotting GC content versus the 6mer toxicity index (*Supplementary file 4*) (*Figure 7E*). While not an absolute requirement, higher GC content made shRNAs more toxic, consistent with reports demonstrating that shRNAs with high GC content in the seed region showed decreased on-target and increased off-target activity (*Gu et al., 2014*; *Ui-Tei et al., 2004*). In summary, our data suggest that si- and/or shRNAs with certain seed sequences are toxic to cancer cells by targeting critical survival genes through an RNAi mechanism independent of both Drosha and Dicer. Furthermore, the data suggest high miRNA content, presumably through competing for occupancy in the RISC, might render cells less sensitive to DISE.

## Discussion

Most current uses of RNAi are aimed toward highly specific silencing with little OTE. In fact, OTEs represent one of the largest impediments to the use of RNAi in phenotypic screening applications. We now demonstrate DISE is a unique form of OTE that results in the simultaneous activation of multiple cell death pathways in cancer cells. The discovery that DISE involves loss of multiple survival genes now provides an explanation for the unique properties we described for this form of cell death, especially the observation that cancer cells have a hard time developing resistance to this cell death mechanism (*Hadji et al., 2014*; *Murmann et al., 2017*).

### DISE represents a specific form of RNAi OTE

There are a number of rules that have been elucidated for designing si/shRNAs (*Bramsen et al., 2009*) to avoid undesired effects such as OTE (*Petri and Meister, 2013*), general toxicity due to the presence of toxic sequence motifs (*Fedorov et al., 2006*; *Petri and Meister, 2013*), poisoning/saturating of the RISC (*Grimm et al., 2006*), or evocation of an IFN response (*Marques and Williams, 2005*). The following arguments and evidence support our prediction that DISE is a manifestation of a novel, functionally important, conserved mechanism of genome regulation, and not the result of one of the above-mentioned effects:

1. The sheer number of toxic shRNAs embedded in CD95L or CD95. A number of genome-wide shRNA and siRNA lethality screens have revealed that 2–5% of shRNAs targeting human genes are toxic to cells. We recently reported in 12 independent arrayed shRNA lethality screens the identification of 651 genes out of about 18,000 targeted genes that are critical for the survival of 8 different cancer cell lines (*Hadji et al., 2014*). Many of the genes targeted by these shRNAs were actually established survival genes (as discussed in [*Hadji et al., 2014*]). That means that the number of shRNAs that are toxic due to a possible OTE or general toxicity would be expected to be very small. In contrast, we found that >80% of the shRNAs and siRNAs that were designed to target either CD95 or CD95L exhibited toxicity in multiple cell lines. Consistent with our data analysis, a parallel genome-scale loss of function screen confirmed that the majority of the tested shRNAs derived from either CD95L and CD95 were toxic to a majority of the tested 216 cell lines when used as a pooled library (*Cowley et al., 2014*). These also included a number of hematopoietic cell lines suggesting that the DISE effect is not limited to solid cancers. Interestingly, in this study the authors did not consider the data on most of the CD95L and CD95 targeting shRNAs to be significant as they received a low consistency score. A high consistency score predicts the observed phenotype (cell death or growth reduction in this case) is caused by knocking down the targeted gene (*Shao et al., 2013*). However, we have demonstrated here that the toxicity of an shRNA is solely dependent on its

seed and the transcriptome of the treated cells. Therefore, the results of every shRNA should be considered individually as far as the DISE inducing effect is concerned.

2. High concentrations of siRNAs can saturate the RISC, preventing the access of crucial endogenous miRNAs (*Khan et al., 2009*). We have demonstrated that, in general, 5 nM of CD95L-derived siRNAs are sufficient to kill cancer cells. We have even seen very efficient cell death with as little as 1 nM of siRNA (see *Figure 2I* and *Figure 1—figure supplement 2E*). It is therefore unlikely we are poisoning the RISC. It has been reported that in siRNA overexpression experiments, changes in mRNA expression can be caused by blocked access of endogenous miRNAs to the RISC, such as the highly expressed miRNA family, let-7 (*Khan et al., 2009*). However, we can exclude such an effect in our analysis, as there was no significant enrichment (or depletion) of the let-7 seed match motif (or that of any other miRNA) in our analyses (black lines in *Figure 5A*).

3. No IFN response was observed. We have performed multiple RNA-Seq and gene array analyses of cells in which DISE was induced by multiple si/shRNAs targeting CD95 or CD95L. In none of these analyses did we detect an increase in any of the known IFN response genes (*Schoggins et al., 2011*) (data not shown). In addition, we demonstrated the latest generation of Dicer optimized 27mer DsiRNAs that do not elicit an IFN response (*Kim et al., 2005*) and the shRNAs expressed from within the cells shown to have low IFN triggering activity (*Robbins et al., 2006*) have the same toxic activities as the standard 21mer siRNAs (see *Figure 1—figure supplement 1A* and *1B*).

4. Mutation of just one position destroys activity. A major argument against DISE toxicity being caused by overloading the RISC, an IFN response or the presence of known toxic sequences, lies in the analysis of the chimeras we generated between siL3 and a non-toxic scrambled oligonucleotide (see *Figure 3H*). This analysis demonstrated that the seed match positions of siL3 are critical for its toxicity. In fact, just replacing one nucleotide in a critical position in the center of the seed match almost completely abolished toxicity of the siRNA.

## What are the requirements for an si/shRNA to induce DISE?

Our data provide strong evidence that the toxicity observed is a sequence-specific event caused by seed matches present in the targets of the toxic si/shRNAs rather than by a toxic motif enriched in all toxic si/shRNAs (i.e. the UGGC motif described before [*Fedorov et al., 2006*]). We did find a correlation between the toxicity of shRNAs (both predicted by the TI and experimentally determined in the shRNA screen) and the GC content in their seed region. While this correlation was significant, it was not a requirement as some of the most toxic si- and shRNAs had a low 8mer seed GC content (shL3, 25%; shR6, 25%; siL3, 37.5%). Our data suggest that survival genes may contain different types of seed matches (based on base composition or sequence) when compared to nonsurvival genes. Such a distinction has indeed been described before (*Stark et al., 2005*). In a study in *Drosophila*, it was determined that survival genes are depleted of seed matches targeted by highly expressed miRNAs. These authors concluded that evolution must have selected against the presence of seed matches for highly expressed miRNAs in the 3'UTR of survival genes. It is therefore not surprising that a gene ontology (GO) analysis of all miRNA targets (the 'targets') in this study described these genes as being involved in development and differentiation (*Stark et al., 2005*). In contrast, genes not targeted by miRNAs (the 'antitargets') grouped in GO clusters that were consistent with cell survival (*Stark et al., 2005*). A similar phenomenon was also shown in mammalian cells; genes with fewer miRNA target sites, as predicted by Targetscan, contained distinct enriched GO terms from those enriched in genes with many predicted target sites. The genes with fewer sites were enriched in GO terms like ribosomal subunits and respiratory chain, whereas target-heavy genes were more enriched in regulatory-related GO terms (*Zare et al., 2014*). It is possible the DISE-inducing si/shRNAs carry seed sequences that preferentially target seed matches present in the 3'UTRs of the 'anti-targets'. However, as our data on the miR-30 based shRNAs suggest, DISE-inducing shRNAs must be expressed at a certain level to be toxic.

## DISE is caused by loading of the guide strand of toxic si/shRNAs into the RISC

Part of our data was generated using a widely used first generation stem loop shRNA platform, the TRC library. The TRC shRNAs have recently been found to be prone to cause OTE. Gu *et al.* showed that the loop design of this system results in imprecise Dicer cleavage and, consequently, the

production of different mature small-RNA species that increase passenger loading, one major source of OTE (*Gu et al., 2012*). More recently, it was reported that most guide RNAs derived from the TRC hairpin were shifted by 4 nt 3' of the expected 5' start site (*Watanabe et al., 2016*). While we did see a shift in processing of these stem loop shRNAs, we did not see such a high level of imprecision in the cleavage of our toxic shRNAs. In fact, 99.4% of the shR6 guide RNAs started at the same nucleotide position (*Figure 5—figure supplement 1A*). The majority of the processing of both our pTIP and pLKO-based shRNAs was shifted by one nucleotide (*Figure 5—figure supplement 1A*). This shift was consistent with the defined seed matches that were detected in the Sylamer analyses. In general, one major seed match was detected with one other minor species (this was less obvious for shL1, *Figure 5—figure supplement 2*). Furthermore, all four Sylamer analyses only detected enrichments in the 3'UTR of downregulated mRNAs that were consistent with only the guide strand targeting the mRNA and not the passenger strand. In all cases, including in cells transfected with the siRNA siL3, the primary enriched sequence motifs were either 7, or 8mers present in the 3'UTR of the targeted mRNAs.

## DISE has features of the RNAi OTE previously reported

Our data on DISE are consistent with a number of properties of RNAi OTE that have previously been reported. Similar to DISE, OTE-mediated silencing requires a 6/7nt seed sequence of complementarity (*Birmingham et al., 2006*; *Jackson et al., 2006*; *Lin et al., 2005*) and it targets mRNAs in the 3'UTR (*Birmingham et al., 2006*). Our data on shRNAs, siRNAs, and DsiRNAs suggest that DISE is not limited to one platform and requires sequence-specific targeting. This conclusion is also consistent with a previous report that suggested that sequence-dependent off-target transcript regulation is independent of the delivery method (*Jackson et al., 2006*). The authors found the same enrichment of 6mers and 7mers in 3'UTRs of targeted mRNAs for siRNAs and shRNAs (*Jackson et al., 2006*).

## The role of Dicer in DISE

We previously reported that Dicer[Exo5-/-] HCT116 cells (with deleted Exon 5) were at least as sensitive to induction of DISE (by either shL3 or shR6) than wt cells suggesting that Dicer deficient cells could be killed by DISE (*Hadji et al., 2014*). It has been reported that these Dicer deficient cells are hypomorphs (*Ting et al., 2008*) and indeed, we detected low residual Dicer expression by western blotting (*Hadji et al., 2014*). We have now revisited this issue with HCT116 cells rendered completely deficient for Dicer using CRISPR/Cas9 gene editing (*Kim et al., 2016*). The fact that these Dicer[-/-] cells were now completely resistant to the toxic effects of shL3 or shR6 demonstrates the complete absence of Dicer protein and activity. Similar to the Drosha[-/-] cells, in the absence of mature miRNAs, which seem to attenuate DISE, Dicer[-/-] cells are hypersensitive to DISE induced by siRNAs.

## Open questions regarding the relevance of DISE

We are proposing an entirely new concept of killing cancer cells that is based on the toxicity of CD95 and CD95L derived small RNAs. Naturally, there are many open questions such as:

1. Is DISE part of an anti-cancer mechanism? We are proposing that DISE kills cancer cells in a way that they usually cannot escape from. We have not found a way to block cancer cells from dying by DISE. We provide strong evidence to suggest this is due to the simultaneous targeting of multiple survival genes that result in the activation of multiple cell death pathways. It will be difficult to prove cells are dying due to the preferential targeting of survival genes. It may never be possible to express multiple siRNA resistant survival genes at the same time at physiological levels to render cancer cells resistant to the action of countless small RNAs. This prediction alone makes DISE a promising new strategy to kill cancer cells.

2. Does CD95L induce DISE *in vivo*? We recently found that overexpression of the CD95L ORF is toxic to cancer cells and that this kills cancer cells in a manner very similar to DISE induction (unpublished data). We and others have noticed upregulation of CD95L in multiple stress-related conditions such as treatment with chemotherapy ([*Friesen et al., 1999*] and data not shown). While the amount of CD95L RNA and the level of upregulation alone may not be enough to induce DISE, it could result from the combined expression of multiple RNAs that when generated kill cells by DISE. We view CD95L as just one of many RNAs that have this activity.

3. Are there other genes in the human genome containing toxic seed sequences? We recently identified other genes in the genome that contain DISE-inducing shRNAs (*Patel and Peter, 2017*). It is therefore possible that when cells are subjected to genotoxic or oncogenic stress that they generate numerous small RNAs that can be taken up by the RISC and in combination execute DISE. Hence, our analysis of CD95/CD95L will likely be applicable to other genes.

## A model for why DISE preferentially kills cancer cells

We interpret the hypersensitivity of both Drosha$^{-/-}$ and Dicer$^{-/-}$ cells to DISE in the following way: Most of the small RNAs in the cells that are loaded into the RISC are miRNAs. Using AGO pull-down experiments, we determined 98.4% of AGO- associated RNAs in HCT116 cells to be miRNAs (99.3% in HeyA8 cells, data not shown). It was recently reported that Drosha$^{-/-}$ cells showed a reduction of miRNA content from roughly 70–80% to 5–6%, and Dicer$^{-/-}$ cells showed a reduction down to 14–21% (*Kim et al., 2016*). Since neither Drosha$^{-/-}$ nor Dicer$^{-/-}$ cells express reduced AGO2 protein levels (see inset in *Figure 3E*), it is reasonable to assume that their RISC can take up many more of the toxic DISE inducing RNAs than the RISC in wt cells explaining the super toxicity of both DISE inducing si/shRNAs and CD95L mRNAs in these cells.

We previously showed expression of either shL3 and shR6 induced DISE in immortalized normal ovarian fibroblasts much more efficiently than in matching nonimmortalized cells (*Hadji et al., 2014*), suggesting that this form of cell death preferentially affects transformed cells. Our data now provide an interesting model to explain the higher sensitivity of cancer cells to DISE when compared to normal cells. It is well documented that cancer cells in general have global downregulation of miRNAs when compared to normal tissues (*Lu et al., 2005*). This might free up the RISC for DISE-inducing RNAs and would imply that miRNAs may protect normal cells from DISE.

Overall, our data allow us to predict that any small RNA with DISE-inducing RNAi activity that does not require Dicer processing can kill cancer cells regardless of Dicer or Drosha status. In fact, in an accompanying manuscript, we demonstrate that DISE can be triggered *in vivo* to treat ovarian cancer in mouse xenografts by delivering CD95L-derived siRNAs using nanoparticles (*Murmann et al., 2017*). No toxicity was observed in the treated mice. These data suggest that it might be possible to develop a novel form of cancer therapy based on the DISE OTE mechanism.

# Materials and methods

**Key resource table**

| Reagent type (species) or resource | Designation | Source or reference | Identifiers | Additional information |
|---|---|---|---|---|
| Gene (Homo sapiens) | CD95L | NA | NM_000639 | |
| Gene (H. sapiens) | CD95 | NA | NM_000043 | |
| Cell line (H. sapiens) | NB7 | PMID: 10802708 | BRENDA Tissue and Enzyme Source Ontology: BTO_0003439; RRID:CVCL_8824 | Human neuroblastoma derived from autonomic ganglia; carries a deletion in both alleles of CASP8 |
| Cell line (H. sapiens) | HeyA8 | PMID: 4016745; PMID: 25984343 | RRID: CVCL_8878; RRID:CVCL_8878 | Human high grade ovarian serous adenocarcinoma; derived from parent Hey cells (RRID: CVCL_0297) |
| Cell line (H. sapiens) | HeyA8 ΔshL3 | this paper | NA | Pool of three HeyA8 cell clones with homozygous 41 nucleotide deletion of the shL3 target site (chr1:172,665,726–172,655,766; Human Dec. 2013 GRCh38/hg38 assembly) produced using CRISPR/Cas9 technology. |
| Cell line (H. sapiens) | HeyA8 ΔsiL3 | this paper | NA | Pool of three HeyA8 cell clones with homozygous 64 nucleotide deletion of the siL3 target site (chr1:172,669,178–172,659,241; Human Dec. 2013 GRCh38/hg38 assembly) produced using CRISPR/Cas9 technology. |

| | | | | |
|---|---|---|---|---|
| Cell line (H. sapiens) | HeyA8 ΔshR6; shR6 k.o. clone #11 | this paper | NA | HeyA8 cell clone #11 with homozygous 227 nucleotide deletion of the shR6 target site (chr10:89,008,920–89,009,146; Human Dec. 2013 GRCh38/hg38 assembly) produced using CRISPR/Cas9 technology; verified homozygous CD95 protein knockout |
| Cell line (H. sapiens) | HeyA8 shR6 k.o. clone #1 | this paper | NA | HeyA8 cell clone #1 with a small deletion and the 227 nucleotide deletion of the shR6 target site and an insertion of the pMJ920 plasmid fragment in CD95 produced using CRISPR/Cas9 technology; verified homozygous CD95 protein knockout |
| Cell line (H. sapiens) | HeyA8 shR6 k.o. clone #2 | this paper | NA | HeyA8 cell clone #2 with a 227 nucleotide deletion of the shR6 target site (chr10:89,008,920–89,009,146; Human Dec. 2013 GRCh38/hg38 assembly) in one allele and an insertion of the pSCB plasmid fragment in the other in CD95 produced using CRISPR/Cas9 technology; verified homozygous CD695 protein knockout |
| Cell line (H. sapiens) | MCF-7 | ATCC | ATCC: HTB-22; RRID:CVCL_0031 | Human adenocarcinoma of the mammary gland, breast; derived from metastatic site: pleural effusion |
| Cell line (H. sapiens) | HCT116 | Korean Collection for Type Cultures (KCTC) | KCTC: cat#HC19023; ATCC: CCL_247; RRID:CVCL_0291 | Human colorectal carcinoma |
| Cell line (H. sapiens) | Drosha$^{-/-}$; Drosha$^{-/-}$ clone #40 | Korean Collection for Type Cultures (KCTC); PMID: 26976605 | KCTC: cat#HC19020 | HCT116 clone #40 with homozygous protein knockout of Drosha; knockout achieved using CRISPR/Cas9 which resulted in a single nucleotide insertion in one allele and a 26 nuceotide deletion in the other |
| Cell line (H. sapiens) | Dicer$^{-/-}$; Dicer$^{-/-}$ clone #43 | Korean Collection for Type Cultures (KCTC); PMID: 26976606 | KCTC: cat#HC19023 | HCT116 clone #43 with homozygous protein knockout of Dicer; knockout achieved using CRISPR/Cas9 which resulted in a three nucleotide insertion and 14 nucleotide deltion in one allele and a 35 nucleotide deletion in the other |
| Cell line (H. sapiens) | Dicer$^{-/-}$; Dicer$^{-/-}$ clone #45 | Korean Collection for Type Cultures (KCTC); PMID: 26976607 | KCTC: cat#HC19024 | HCT116 clone #45 with homozygous protein knockout of Dicer; knockout achieved using CRISPR/Cas9 which resulted in a 53 nucleotide deltion in one allele and a 28 nuceotide deletion in the other |
| Cell line (H. sapiens) | 293T | ATCC | ATCC: CRL-3216; RRID:CVCL_0063 | Derived from HEK293 cells (ATCC: CRL-1573); express large T antigen; used for packaging viruses |
| Cell line (H. sapiens) | 293T ΔshL3 | this paper | NA | Pool of three 293T cell clones with homozygous 41 nucleotide deletion of the shL3 target site (chr1:172,665,726–172,655,766; Human Dec. 2013 GRCh38/hg38 assembly) produced using CRISPR/Cas9 technology. |
| Cell line (H. sapiens) | Phoenix-AMPHO | ATCC | ATCC: CRL-3213; RRID:CVCL_H716 | Second generation retrovirus producer cell line |
| Antibody | anti-β-actin antibody (mouse monoclonal) | Santa Cruz | Santa Cruz: cat#sc-47778; RRID:AB_626632 | 1:2000; for western blot; primary Ab |
| Antibody | anti-human CD95L (Mouse IgG1 monoclonal) | BD Biosciences | BD Biosciences: cat#556387; RRID:AB_396402 | 1:500; for western blot; primary Ab |

| Antibody | anti-human CD95 (rabbit polyclonal) | Santa Cruz | Santa Cruz: cat#sc-715; RRID:AB_2100386 | 1:1000; for western blot; primary Ab |
|---|---|---|---|---|
| Antibody | anti-human AGO2 (rabbit monoclonal) | Abcam | Abcam: cat#AB186733; RRID:AB_2713978 | 1:2000; for western blot; primary Ab |
| Antibody | anti-human Drosha (rabbit monoclonal) | Cell Signaling | Cell Signaling: cat#3364; RRID:AB_2238644 | 1:1000; for western blot; primary Ab |
| Antibody | anti-human Dicer (rabbit polyclonal) | Cell Signaling | Cell Signaling: cat#3363; RRID:AB_2093073 | 1:1000; for western blot; primary Ab |
| Antibody | Goat anti-rabbit, IgG-HRP | Southern Biotech | Southern Biotech: cat#SB-4030–05; RRID:AB_2687483 | 1:5000; for western blot; secondary Ab |
| Antibody | Goat anti-rabbit, IgG-HRP | Cell Signaling | Cell Signaling: cat#7074; RRID:AB_2099233 | 1:2000; for western blot; secondary Ab |
| Antibody | Goat anti-mouse; IgG1-HRP | Southern Biotech | Southern BioTech: cat#1070–05; RRID:AB_2650509 | 1:5000; for western blot; secondary Ab |
| Isotype control | FITC-mouse IgG1, κ isotype control | BD Biosciences | BD Biosciences: cat#551954; RRID:AB_394297 | 4 uL used for 1 × 106 cells; for flow cytometry |
| Antibody | FITC-mouse anti-Human CD95 | BD Biosciences | BD Biosciences: cat#556640; RRID:AB_396506 | 4 uL used for 1 × 106 cells; for flow cytometry |
| Recombinant protein reagent | sCD95L (S2) | PMID: 14504390 | NA | Soluble form of human CD95L (amino acids 137–281); recombinant protein |
| Recombinant protein reagent | LzCD95L | PMID: 14504390 | NA | Leucine zipper tagged CD95L; recombinant protein |
| Chemical compound | propidium iodide | Sigma-Aldrich | Sigma-Aldrich: cat#P4864 | Used for subG1 flow cytometry analysis |
| Chemical compound | puromycin | Sigma-Aldrich | Sigma-Aldrich: cat#P9620 | Used for selection of cells expressing puromycin resistance cassettes |
| Chemical compound | G418 | Affymetrix | Affymetrix: cat#11379 | Used for selection of cells expressing G418 resistance cassette |
| Recombinant DNA reagent | venus-CD95L sensor (plasmid) | this paper | NA | Modified CD510B-1 lentiviral vector (PMID: 25366259) was used as backbone; vector expresses a venus-human CD95L conjugate mRNA that can be used to monitor RNAi activity of si/shRNAs targeting CD95L using venus fluorescence. |
| Recombinant DNA reagent | venus-CD95 sensor (plasmid) | this paper | NA | Modified CD510B-1 lentiviral vector (PMID: 25366259) was used as backbone; vector expresses a venus-human CD95 conjugate mRNA that can be used to monitor RNAi activity of si/shRNAs targeting CD95 using venus fluorescence. |
| Recombinant DNA reagent | pLenti-GIII-CMV-RFP-2A-Puro vector; pLenti | ABM Inc | NA | pLenti control empty lentiviral vector; carries an RFP-2a-puromycin resistance cassette |
| Recombinant DNA reagent | pLNCX2 | Clontech | Clontech: cat#631503 | pLNCX2 control empty retroviral vector; carries a neomycin resistance cassette |

| Recombinant DNA reagent | pTIP | PMID: 24656822 | NA | Lentivirus used for doxycycline-induced expression of shRNAs; contains puromycin resistance cassette; modified from the original backbone which contained a GFP cassette instead of a puromycin cassette (PMID: 17311008); original backbone from the Rossi lab. |
|---|---|---|---|---|
| Recombinant DNA reagent | pLenti-CD95L-WT | this paper | NA | pLenti-GIII-CMV-RFP-2A-Puro vector that expresses human wild type CD95L cDNA (NM_000639.2); used to express wt human CD95L upon infection with lentiviral particles |
| Recombinant DNA reagent | pLenti-CD95L-L1MUT | this paper | NA | pLenti-GIII-CMV-RFP-2A-Puro vector that expresses human CD95L cDNA (NM_000639.2) with 8 silentmutations overlapping the shL1 target site (GCATCATCTTTGGAGAAGCAA - > GCCTCGTCCCTAGAAAAACAG); used to express shL1-resistant human CD95L upon infection with lentiviral particles |
| Recombinant DNA reagent | pLenti-CD95L-L3MUT | this paper | NA | pLenti-GIII-CMV-RFP-2A-Puro vector that expresses human CD95L cDNA (NM_000639.2) with 8 silent mutations overlapping the shL3 target site (ACTGGGCTGTACTTTGTATAT - > ACCGGATTATATTTCGTGTAC); used to express shL3-resistant human CD95L upon infection with lentiviral particles |
| Recombinant DNA reagent | pLNCX2-CD95-WT | this paper | NA | pLNCX2 vector that expresses human CD95 cDNA (BC012479.1); used to express wild type CD95 upon infection with lentiviral particles |
| Recombinant DNA reagent | pLNCX2-CD95-R6MUT | this paper | NA | pLNCX2 vector that expresses mutant human CD95 cDNA (BC012479.1) which contains 8 silent mutations overlapping the shR6 site (GTGTCGCTGTAAACCAAACTT - > ATGTCGCTGCAAGCCCAATTT); used to express shR6-resistant CD95 upon infection with lentiviral particles |
| Transfected construct | gRNA scaffold | PMID: 23287722 | IDT: synthesized as gene block | 455 nucleotide CRISPR/Cas9 gRNA scaffold synthesized as a gene block; contains promoter, gRNA scaffold, target sequence, and termination sequence; scaffold transcribes gRNAs that target Cas9 endonuclease to cut at target sites; target sequences consist of 19 nucleotides that are complementary to the target site of choice; co-transfected with Cas9 to catalyze cleavage. |
| Transfected construct | pMJ920 Cas9 plasmid | Addgene; PMID: 23386978 | Addgene: cat#42234 | Plasmid that expresses a human codon-optimized Cas9 tagged with GFP and HA; used to express Cas9 for CRISPR-mediated deletions. |
| Chemical compound | Lipofectamine 2000 | ThermoFisher Scientific | ThermoFisher Scientific: cat#11668019 | Transfection reagent |
| Chemical compound | Lipofectamine RNAiMAX | ThermoFisher Scientific | ThermoFisher Scientific: cat#13778150 | Transfection reagent; used for transfection of small RNAs such as siRNAs |

| Commercial assay or kit | StrataClone Blunt PCR Cloning Kit | Agilent Technologies | Agilent Technologies: cat#240207 | Used to blunt-end clone the gRNA scaffolds into the pSC-B plasmid |
|---|---|---|---|---|
| Commercial assay or kit | High-Capacity cDNA reverse transcription kit | Applied Biosystems | 4368814 | |
| Array cards preloaded with primers | 384-well TLDA cards | Applied Biosystems | 43422489 | |
| Commercial assay kit | Taqman Gene expression master mix | ThermoFisher Scientific | 4369016 | |
| Sequence-based reagent | shL3 flanking Fr primer | IDT | IDT: custom DNA oligo | Fr primer that flanks shL3 site; used to detect 41 nt shL3 deletion; 5'-TCTGGAATGGGAAGACACCT-3' |
| Sequence-based reagent | shL3 flanking Rev primer | IDT | IDT: custom DNA oligo | Rev primer that flanks shL3 site; used to detect 41 nt shL3 deletion; 5'-CCTCCATCATCACCAGATCC-3' |
| Sequence-based reagent | shL3 internal Rev primer | IDT | IDT: custom DNA oligo | Rev primer that overlaps with the shL3 site; used to detect 41 nt shL3 deletion; 5'-ATATACAAAGTACAGCCCAGT-3' |
| Sequence-based reagent | shR6 flanking Fr primer | IDT | IDT: custom DNA oligo | Fr primer that flanks shR6 site; used to detect 227 nt shR6 deletion; 5'-GGTGTCATGCTGTGACTGTTG-3' |
| Sequence-based reagent | shR6 flanking Rev primer | IDT | IDT: custom DNA oligo | Rev primer that flanks shR6 site; used to detect 227 nt shR6 deletion; 5'-TTTAGCTTAAGTGGCCAGCAA-3' |
| Sequence-based reagent | shR6 internal Rev primer | IDT | IDT: custom DNA oligo | Rev primer that overlaps with the shR6 site; used to detect 227 nt shR6 deletion; 5'-AAGTTGGTTTACATCTGCAC-3' |
| Sequence-based reagent | siL3 flanking Fr primer | IDT | IDT: custom DNA oligo | Fr primer that flanks siL3 site; used to detect 64 nt siL3 deletion; 5'-CTTGAGCAGTCAGCAACAGG-3' |
| Sequence-based reagent | siL3 flanking Rev primer | IDT | IDT: custom DNA oligo | Rev primer that flanks siL3 site; used to detect 64 nt siL3 deletion; 5'-CAGAGGTTGGACAGGGAAGA-3' |
| Sequence-based reagent | siL3 internal Rev primer | IDT | IDT: custom DNA oligo | Rev primer that is internal to the siL3 site; used to detect 64 nt siL3 deletion; 5'-ATATGGGTAATTGAAGGGCTG-3'. |
| Sequence-based reagent | siScr | IDT; Dharmacon | Dharmacon #D-001810-02-05 | sense: UGGUUUACAUGUUGUGUGA |
| Sequence-based reagent | siL1 | Dharmacon | L-011130-00-0005 | sense: UACCAGUGCUGAUCAUUUA |
| Sequence-based reagent | siL1 | IDT | customer synthesis | sense: UACCAGUGCUGAUCAUUUA |
| Sequence-based reagent | siL2 | IDT | customer synthesis | sense: CAACGUAUCUGAGCUCUCU |
| Sequence-based reagent | siL3 | IDT | customer synthesis | sense: GCCCUUCAAUUACCCAUAU |
| Sequence-based reagent | siL3MUT | IDT | IDT #51-01-14-03 | sense: GGACUUCAACUAGACAUCU |
| Sequence-based reagent | siL4 | IDT | customer synthesis | sense: GGAAAGUGGCCCAUUUAAC |
| Sequence-based reagent | shL3 => siL3 | IDT | customer synthesis | sense: GACUGGGCUGU ACUUUGUAdTdA antisense: UACAAAGUACA GCCCAGUUdTdT |
| Sequence-based reagent | shR6 => siR6 | IDT | customer synthesis | sense: GGGUGCAGAU GUAAACCAAAdCdT; antisense: UUUGGUUUACA UCUGCACUUdTdT |
| Sequence-based reagent | Dsi-13.2 | IDT | customer synthesis | sense: AUCUU ACCAGUGC UGAUCAUUUAdTdA |
| Sequence-based reagent | Dsi-13.3 | IDT | customer synthesis | sense: AAAGUAUACUU CCGGGGUCAAUCdTdT |
| Sequence-based reagent | Dsi-13.9 | IDT | customer synthesis | sense: CUUCCGGGG UCAAUCUUGCAACAdAdC |
| Sequence-based reagent | Dsi-13.x | IDT | customer synthesis | sense: CAGGACUGAGAAG AAGUAAAACCdGdT |

| Sequence-based reagent | DsiL3 | IDT | customer synthesis | sense: CAGCCCUUCAAU UACCCAUAUCCdCdC |
|---|---|---|---|---|
| Sequence-based reagent | siScr pool | Dharmacon | D-001810–10 | |
| Sequence-based reagent | smartpool siRNA targeting NUCKS1 | Dharmacon | L-014208–02 | |
| Sequence-based reagent | smartpool siRNA targeting CAPZA1 | Dharmacon | L-012212–00 | |
| Sequence-based reagent | smartpool siRNA targeting CCT3 | Dharmacon | L-018339–00 | |
| Sequence-based reagent | smartpool siRNA targeting FSTL1 | Dharmacon | L-013615–00 | |
| Sequence-based reagent | smartpool siRNA targeting FUBP1 | Dharmacon | L-011548–00 | |
| Sequence-based reagent | smartpool siRNA targeting GNB1 | Dharmacon | L-017242–00 | |
| Sequence-based reagent | smartpool siRNA targeting NAA50 | Dharmacon | L-014597–01 | |
| Sequence-based reagent | smartpool siRNA targeting PRELID3B | Dharmacon | L-020893–01 | |
| Sequence-based reagent | smartpool siRNA targeting SNRPE | Dharmacon | L-019719–02 | |
| Sequence-based reagent | smartpool siRNA targeting TFRC | Dharmacon | L-003941–00 | |
| Sequence-based reagent | smartpool siRNA targeting HIST1H1C | Dharmacon | L-006630–00 | |
| Sequence based reagent (human) | GAPDH primer | Thermofisher Scientific | Hs00266705_g1 | |
| Sequence based reagent (human) | CD95 primer | Thermofisher Scientific | custom probe | Fr primer: GGCTAACCCC ACTCTATGAATCAAT Rev primer: GGCCTGCCT GTTCAGTAACT Probe: CCTT TTGCTGAAATATC |
| Sequence based reagent (human) | CD95 primer (*Figure 5—figure supplement 3*) | Thermofisher Scientific | Hs00163653_m1 | |
| Sequence based reagent (human) | CD95L primers | Thermofisher Scientific | Hs00181226_g1; Hs00181225_m1 | |
| Sequence based reagent (human) | shL3 target site in CD95L | Thermofisher Scientific | custom probe | Fr primer: *GGTGGCC TTGTGATCAATGAAA* Rev primer: *GCAAGA TTGACCCCGGAAGTATA* Probe: *CTG GGCTGTACTTTGTATATT* |
| Sequence based reagent (human) | downstream of shL3 site | Thermofisher Scientific | custom probe | Fr primer: *CCCC AGGATCTGGTGATGATG* Rev primer: *ACTG CCCCCAGGTAGCT* Probe: *CCCAC ATCTGCCCAGTAGT* |
| Sequence based reagent (human) | GAPDH primer (TLDA card) | Thermofisher Scientific | Hs99999905_m1 | |
| Sequence based reagent (human) | ATP13A3 primer (TLDA card) | Thermofisher Scientific | Hs00225950_m1 | |
| Sequence based reagent (human) | CAPZA1 primer (TLDA card) | Thermofisher Scientific | Hs00855355_g1 | |
| Sequence based reagent (human) | CCT3 primer (TLDA card) | Thermofisher Scientific | Hs00195623_m1 | |
| Sequence based reagent (human) | FSTL1 primer (TLDA card) | Thermofisher Scientific | Hs00907496_m1 | |

| | | | | |
|---|---|---|---|---|
| Sequence based reagent (human) | FUPB1 primer ( TLDA card) | Thermofisher Scientific | Hs00900762_m1 | |
| Sequence based reagent (human) | GNB1 primer (TLDA card) | Thermofisher Scientific | Hs00929799_m1 | |
| Sequence based reagent (human) | HIST1H1C primer (TLDA card) | Thermofisher Scientific | Hs00271185_s1 | |
| Sequence based reagent (human) | NAA50 primer (TLDA card) | Thermofisher Scientific | Hs00363889_m1 | |
| Sequence based reagent (human) | NUCKS1 primer (TLDA card) | Thermofisher Scientific | Hs01068059_g1 | |
| Sequence based reagent (human) | PRELID3B primer (TLDA card) | Thermofisher Scientific | Hs00429845_m1 | |
| Sequence based reagent (human) | SNRPE primer (TLDA card) | Thermofisher Scientific | Hs01635040_s1 | |
| Sequence based reagent (human) | TFRC primer (TLDA card) | Thermofisher Scientific | Hs00951083_m1 | |
| Software | Stata 14 | Stata | | RRID:SCR_012763 |
| Software | Rstudio (R3.3.1) | Rstudio | | RRID:SCR_000432 |
| sequence based reagent | shScr | Sigma | SHC002V | Non-targeting shRNA control transduction particles |
| sequence based reagent (human) | shL1 | Sigma | TRCN0000058998 | GCATCATCTTTGGAGAAGCAA |
| sequence based reagent (human) | shL2 | Sigma | TRCN0000058999 | CCCATTTAACAGGCAAGTCCA |
| sequence based reagent (human) | shL3 | Sigma | TRCN0000059000 | ACTGGGCTGTACTTTGTATAT |
| sequence based reagent (human) | shL4 | Sigma | TRCN0000059001 | GCAGTGTTCAATCTTACCAGT |
| sequence based reagent (human) | shL5 | Sigma | TRCN0000059002 | CTGTGTCTCCTTGTGATGTTT |
| sequence based reagent (human) | shL6 | Sigma | TRCN0000372231 | TGAGCTCTCTCTGGTCAATTT |
| sequence based reagent (human) | shL2' | Sigma | TRCN0000372232 | TAGCTCCTCAACTCACCTAAT |
| sequence based reagent (human) | shL5' | Sigma | TRCN0000372175 | GACTAGAGGCTTGCATAATAA |
| sequence based reagent (human) | shR2 | Sigma | TRCN0000218492 | CTATCATCCTCAAGGACATTA |
| sequence based reagent (human) | shR5 | Sigma | TRCN0000038695 | GTTGCTAGATTATCGTCCAAA |
| sequence based reagent (human) | shR6 | Sigma | TRCN0000038696 | GTGCAGATGTAAACCAAACTT |
| sequence based reagent (human) | shR7 | Sigma | TRCN0000038697 | CCTGAAACAGTGGCAATAAAT |
| sequence based reagent (human) | shR8 | Sigma | TRCN0000038698 | GCAAAGAGGAAGGATCCAGAT |
| sequence based reagent (human) | shR27' | Sigma | TRCN0000265627 | TTTTACTGGGTACATTTTATC |
| sequence based reagent (human) | shR7' | Sigma | TRCN0000255407 | TTAAATTATAATGTTTGACTA |
| sequence based reagent (human) | shR8' | Sigma | TRCN0000255408 | ATATCTTTGAAAGTTTGTATT |
| sequence based reagent (human) | shR6' | Sigma | TRCN0000255406 | CCCTTGTGTTTGGAATTATAA |

## Reagents and antibodies

Primary antibodies for Western blot: anti-β-actin antibody (Santa Cruz #sc-47778, RRID:AB_626632), anti-human CD95L (BD Biosciences #556387, RRID:AB_396402), and anti-human CD95 (Santa Cruz #sc-715, RRID:AB_2100386), anti-human AGO2 (Abcam #AB186733, RRID:AB_2713978), anti-human Drosha (Cell Signaling #3364, RRID:AB_2238644), and anti-Dicer (Cell Signaling #3363, RRID:AB_2093073). Secondary antibodies for Western blot: Goat anti-rabbit; IgG-HRP (Southern Biotech #SB-4030–05, RRID:AB_2687483 and Cell Signaling #7074, RRID:AB_2099233) and Goat anti-mouse; IgG1-HRP; (Southern BioTech #1070–05, RRID:AB_2650509). Conjugated antibody isotype control for CD95 surface staining were FITC-mouse anti-human CD95 (BD Biosciences #556640, RRID:AB_396506) and FITC-mouse IgG1, K isotype control (BD Biosciences #551954, RRID:AB_394297). Recombinant soluble S2 CD95L and leucine-zipper tagged (Lz)CD95L were described before (*Algeciras-Schimnich et al., 2003*). Reagents used: propidium iodide (Sigma-Aldrich #P4864), puromycin (Sigma-Aldrich #P9620), G418 (Affymetrix #11379), doxycycline (Dox) (Sigma-Aldrich #9891), Lipofectamine 2000 (ThermoFisher Scientific #11668027), and Lipofectamine RNAiMAX (ThermoFisher Scientific #13778150).

## Cell lines

The ovarian cancer cell line HeyA8 (RRID:CVCL_8878), the neuroblastoma cell line NB7 (RRID:CVCL_8824), and the breast cancer cell line MCF-7 (RRID:CVCL_0031) were grown in RPMI 1640 medium (Cellgro #10–040 CM), 10% heat-inactivated FBS (Sigma-Aldrich), 1% L-glutamine (Mediatech Inc), and 1% penicillin/streptomycin (Mediatech Inc). The human embryonic kidney cell line 293T (RRID:CVCL_0063) and Phoenix AMPHO (RRID:CVCL_H716) cells were cultured in DMEM (Cellgro #10–013 CM), 10% heat-inactivated FBS, 1% L-Glutamine, and 1% penicillin/streptomycin.

HCT116 Drosha$^{-/-}$ and Dicer$^{-/-}$ cells were generated by Narry Kim (*Kim et al., 2016*). HCT116 parental (cat#HC19023, RRID:CVCL_0291), a Drosha$^{-/-}$ clone (clone #40, cat#HC19020) and two Dicer$^{-/-}$ clones (clone #43, cat#HC19023 and clone #45, cat#HC19024) were purchased from Korean Collection for Type Cultures (KCTC). All HCT116 cells were cultured in McCoy's medium (ATCC, cat#30–2007), 10% heat-inactivated FBS, 1% L-Glutamine, and 1% penicillin/streptomycin. All cell lines were authenticated using STR profiling and tested monthly for mycoplasm using PlasmoTest (Invitrogen).

All lentiviruses were generated in 293T cells using pCMV-dR8.9 and pMD.G packaging plasmids. Retroviruses were generated in Phoenix AMPHO cells using the VSVg packaging plasmid.

NB7 cells overexpressing wild type and mutant CD95L cDNAs used in *Figure 1C and D* were generated by infecting cells seeded at 50,000 to 100,000 cells per well on a 6-well plate with empty pLenti, pLenti-CD95L-WT, pLenti-CD95L-L1MUT, and pLenti-CD95L-L3MUT (described below) with 8 μg/ml polybrene. Selection was done with 3 μg/ml puromycin for at least 48 hr.

MCF-7 cells overexpressing CD95 cDNAs used in *Figure 1F* were generated by seeding cells at 50,000 per well in a 6-well plate followed by infection with pLNCX2-CD95 or pLNCX2-CD95R6MUT (described below) in the presence of 8 μg/ml polybrene. Selection was done with 200 μg/ml G418 48 hrs after infection for 2 weeks.

The HeyA8 cells used in *Figure 3D* carried a lentiviral Venus-siL3 sensor vector (*Murmann et al., 2017*) and were infected with NucLight Red lentivirus (Essen Bioscience #4476) with 8 μg/ml polybrene and selected with 3 μg/ml puromycin and sorted for high Venus expression 48 hr later. HeyA8 ΔshR6 clone #2 sensor cells used in *Figure 3A–3C* were infected with lentiviruses generated from the Venus-CD95L sensor vector (described below) to over-express the Venus-CD95L chimeric transcript. Cells were sorted for high Venus expression 48 hr later. NB7 cells over-expressing either the Venus-CD95L sensor or the Venus-CD95 sensor (described below) used in *Figure 6—figure supplement 1A* were similarly generated.

## Plasmids and constructs

The Venus-CD95L ORF and Venus-CD95 ORF (full length) sensor vectors were created by sub-cloning the Venus-CD95L or the Venus-CD95 inserts (synthesized as a minigene by IDT with flanking XbaI RE site on the 5' end and EcoRI RE site at the 3' end in the pIDTblue vector), which are composed of the Venus ORF followed by either the CD95L ORF (accession number NM_000639.2) or the CD95 ORF (accession number BC012479.1) as an artificial 3'UTR (both lacking the A in the start

codon), respectively, into the modified CD510B vector (*Ceppi et al., 2014*) using XbaI and EcoRI. Ligation was done with T4 DNA ligase.

The pLNCX2-CD95R6MUT vector was synthesized by replacing a 403 bp fragment of the CD95 ORF insert from the pLNCX2-CD95-WT vector (*Hadji et al., 2014*) with a corresponding 403 bp fragment that had eight silent mutation substitutions at the shR6 site (5'-*GTGTCGCTGTAAACCAAACTT* - > 5'-*ATGTCGCTGCAAGCCCAATTT*-3') using BstXI (NEB #R0113) and BamHI (NEB #R3136) restriction enzymes (mutant insert was synthesized in a pIDTblue vector with 5' end BstXI site and 3' end BamHI RE site).

Dox-inducible vectors expressing shRNAs were constructed by subcloning an annealed double-stranded DNA insert containing the sequence encoding the shRNA hairpin (sense strand: 5'-*TGGCTTTATATATCTCCCTATCAGTGATAGAGATCGNNNNNNNNNNNNNNNNNNNNNCTCGAG nnnnnnnnnnnnnnnnnnnnnTTTTTGTACCGAGCTCGGATCCACTAGTCCAGTGTGGGCATGCTGCG TTGACATTGATT*-3') into the pTIP-shR6 vector (*Hadji et al., 2014*). BsaBI (NEB #R0537) and SphI-HF (NEB #R3182) were used to digest both the pTIP-shR6 vector (to excise the shR6 insert) and the double-stranded shRNA DNA cassette insert followed by ligation with T4 DNA ligase. The template oligos were purchased from IDT. The poly-N represents the two 21 bp sequences that transcribe for the sense (*N*) and antisense (*n*) shRNA. miR-30 based shRNAs were generated by The Gene Editing and Screening Core, at Memorial Sloan Kettering, NY, by converting the 21mers expressed in the pLKO and pTIP vectors into 22mers followed by cloning into the Dox-inducible LT3REPIR vector as described (*Dow et al., 2012*). A vector expressing an shRNA against Renilla luciferase was used as control (*Dow et al., 2012*).

## CRISPR deletions

We identified two gRNAs that target upstream and downstream of the site to be deleted. These gRNAs were expected to result in the deletion of a DNA piece just large enough to remove the target site. The CRISPR gRNA scaffold gene blocks were from IDT and consisted of the DNA sequence 5'-*TGTACAAAAAAGCAGGCTTTAAAGGAACCAATTCAGTCGACTGGATCCGGTACCAAGG TCGGGCAGGAAGAGGGCCTATTTCCCATGATTCCTTCATATTTGCATATACGATACAAGGCTG TTAGAGAGATAATTAGAATTAATTTGACTGTAAACACAAAGATATTAGTACAAAATACGTGACG TAGAAAGTAATAATTTCTTGGGTAGTTTGCAGTTTTAAAATTATGTTTTAAAATGGACTATCATATGC TTACCGTAACTTGAAAGTATTTCGATTTCTTGGCTTTATATATCTTGTGGAAAGGACGAAACACCG NNNNNNNNNNNNNNNNNNNGTTTTAGAGCTAGAAATAGCAAGTTAAAATAAGGCTAGTCCG TTATCAACTTGAAAAAGTGGCACCGAGTCGGTGCTTTTTTTCTAGACCCAGCTTTCTTGTACAAAG TTGGCATTA*-3' (*Mali et al., 2013*); The poly-*NNNNNNNNNNNNNNNNNNNN* represents the 19nt target sequence. The two 19nt target sequences for excision of the shL3 site (Δ41 deletion) were 5'-*CCTTGTGATCAATGAAACT*-3' (gRNA #1) and 5'-*GTTGTTGCAAGATTGACCC*-3' (gRNA #2). The two target sequences for the Δ227 deletion of the shR6 site were 5'-*GCACTTGGTATTCTGGGTC*-3' and 5'-*TGTTTGCTCATTTAAACAC*-3'. The two target sequences for Δ64 deletion of the siL3 site were 5'-*TAAAACCGTTTGCTGGGGC*-3' and 5'-*TATCCCCAGATCTACTGGG*-3'. Target sequences were identified using the CRISPR gRNA algorithm found at http://crispr.mit.edu/; only gRNAs with scores over 50 were used. These six gene blocks were sub-cloned into the pSC-B-amp/kan plasmid using the StrataClone Blunt PCR Cloning kit (Agilent Technologies #240207).

The target sites of siL3, shL3, and shR6 were homozygously deleted from target cells by co-transfecting Cas9 plasmid with each corresponding pair of pSC-B-gRNA plasmids. Briefly, 400,000 cells were seeded per well on a 6-well plate the day prior to transfection. Each well was transfected with 940 ng of Cas9-GFP plasmid (pMJ920) (*Jinek et al., 2013*) and 450 ng of each pSC-B-gRNA plasmid using Lipofectamine 2000. Media was replaced next day. One to two days later, cells were sorted for the top 50% population with the highest green fluorescence. Those cells were cultured for an additional week to let them recover. The cells were then sorted by FACS (BD FACSAria SORP system) directly into 96-well plates containing a 1:1 ratio of fresh media:conditioned media for single cell cloning. Approximately two to three weeks later, single cell clones were expanded and subjected to genotyping. PCR using both a primer pair that flanked the region to be deleted and another pair containing one flanking primer and one internal primer was used to screen clones for homozygous deletion. For detection of the Δ41 deletion of the shL3 site, the flanking external primers were 5'-*TCTGGAATGGGAAGACACCT*-3' (Fr primer) and 5'- *CCTCCATCATCACCAGATCC*-3' (Rev primer), and the internal Rev primer was 5'-*ATATACAAAGTACAGCCCAGT*-3'. For detection of

the Δ227 deletion of the shR6 site, the flanking external primers were 5'-*GGTGTCATGCTGTGACTG TTG*-3' (Fr primer) and 5'-*TTTAGCTTAAGTGGCCAGCAA*-3' (Rev primer), and the internal Rev primer was 5'-*AAGTTGGTTTACATCTGCAC*-3'. For detection of the Δ64 deletion of the siL3 site, the flanking external primers were 5'-*CTTGAGCAGTCAGCAACAGG*-3' (Fr primer) and 5'-*CAGAGG TTGGACAGGGAAGA*-3' (Rev primer), and the internal Rev primer was 5'-*ATATGGGTAA TTGAAGGGCTG*-3'. After screening the clones, Sanger sequencing was performed to confirm that the proper deletion had occurred. Three clones were pooled for each si/shRNA target site deletion except for HeyA8 ΔshR6 for which only clone #11 showed homozygous deletion of the shR6 site; clones #1 and 2 were not complete shR6 deletion mutants, but frame-shift mutations did occur in each allele (as in clone #11) making them CD95 knockout clones as depicted in *Figure 2—figure supplement 1A and B*.

## Knockdown with pLKO lentiviruses

Cells were infected with the following pLKO.2 MISSION Lentiviral Transduction Particles (Sigma): pLKO.2-puro non-targeting (scramble) shRNA particles (#SHC002V), eight non-overlapping shRNAs against human CD95L mRNA (accession number #NM_000639), TRCN0000058998 (shL1: GCATCA TCTTTGGAGAAGCAA), TRCN0000058999 (shL2: CCCATTTAACAGGCAAGTCCA), TRCN0000059000 (shL3: ACTGGGCTGTACTTTGTATAT), TRCN0000059001 (shL4: GCAGTGTTCAA TCTTACCAGT), TRCN0000059002 (shL5: CTGTGTCTCCTTGTGATGTTT), TRCN0000372231 (shL6: TGAGCTCTCTCTGGTCAATTT), TRCN0000372232 (shL2': TAGCTCCTCAACTC ACCTAAT), and TRCN0000372175 (shL5': GACTAGAGGCTTGCATAATAA), and nine non-overlapping shRNAs against human CD95 mRNA (accession number NM_000043), TRCN0000218492 (shR2: CTATCA TCCTCAAGGACATTA), TRCN00000 38695 (shR5: GTTGCTAGATTATCGTCCAAA), TRCN0000038696 (shR6: GTGCAGA TGTAAACCAAACTT), TRCN0000038697 (shR7: CCTGAAA-CAGTGGCAATAAAT), TRCN0000038698 (shR8: GCAAAGAGGAAGGATCCAGAT), TRCN0000265627 (shR27': TTTTACTGGGTACATTTATC), TRCN0000255406 (shR6': CCCTTGTGTTT GGAATTATAA), TRCN0000255407 (shR7': TTAAATTATAATGTTTGACTA), and TRCN0000255408 (shR8': ATATCTTTGAAAGTTTGTATT). Infection was carried out according to the manufacturer's protocol. In brief, 50,000 to 100,000 cells seeded the day before in a 6-well plate were infected with each lentivirus at an M.O.I of 3 in the presence of 8 µg/ml polybrene overnight. Media change was done the next day, followed by selection with 3 µg/ml puromycin 24 hrs later. Selection was done for at least 48 hrs until puromycin killed the non-infected control cells. For infection of NB7 cells over-expressing pLenti-CD95L cDNAs with pLKO lentiviral particles as in *Figure 1C and D*, cells were seeded at 5000 per well on a 24-well plate and infected with an M.O.I. of 20 to ensure complete infection. For infection of MCF-7 cells over-expressing pLNCX2-CD95 cDNAs with pLKO lentiviruses as in *Figure 1G*, cells were seeded at 7000 per well on a 24-well plate and infected at an M.O.I. of three. 3 µg/ml puromycin was added 48 hrs after infection. For infection of HCT116, Drosha$^{-/-}$, and Dicer$^{-/-}$ cells in *Figure 3E*, cells were seeded at 100,000 per well in a 24-well plate and infected at an M.O.I of three. 3 µg/ml puromycin was added 48 hrs after infection.

## Knockdown with pTIP-shRNA viruses

Cells were plated at 50,000 to 100,000 cells per well in a 6-well plate. Cells were infected with lentivirus generated in 293T cells from the desired pTIP-shRNA vector in the presence of 8 µg/ml Polybrene. Media was replaced 24 hrs later. Selection was done 48 hrs after infection with 3 µg/ml puromycin. Induction of shRNA expression was achieved by adding 100 ng/ml Dox to the media. For infection with the LT3REPIR-shRNA viruses cells were plated and infected as described above for pTIP-shRNA viruses. After selection with 3 µg/ml puromycin was complete, they were plated in 96-well plates and the shRNA expression was induced by adding Dox (100 ng/ml) to the media. The cell confluency over time was measured using Incucyte.

## Transfection with short oligonucleotides

siRNAs were either purchased from Dharmacon (*Figures 2I* and *4D*, *Figure 1—figure supplement 1A*, *Figure 5—figure supplement 2*) or synthesized by IDT (*Figure 3* and *Figure 5—figure supplement 1B*) as sense and antisense RNA (or DNA for *Figure 3B*) oligos and annealed. The sense RNA oligonucleotides had 3' two deoxy-T overhangs. The antisense RNA oligos were phosphorylated at

the 5' end and had 3' two deoxy-A overhangs. siRNAs targeting CD95L (and controls) were as follows: siRNA (Scr, sense: UGGUUUACAUGUUGUGUGA), siL1 (sense: UACCAGUGCUGAUCAUUUA), siL2 (sense: CAACGUAUCUGAGCUCUCU), siL3 (sense: GCCCUUCAAUUACCCAUAU), siL4 (sense: GGAAAGUGGCCCAUUUAAC), and siL3MUT (sense: GGACUUCAACUAGACAUCU). The siL3 DNA oligos (sense: GCCCTTCAATTACCCATAT) and Scr DNA oligos (sense: TGGTTTACATGTTGTGTGA) were used in *Figure 3B*. Blunt siL3 and siScr RNA oligos without the deoxynucleotide overhangs as well as siL2 and siL3 RNA oligos with Cy5-labelled 5' or 3' ends (IDT) were used in *Figure 3C*. DsiRNA used in *Figure 1—figure supplement 1* were Dsi13.X (sense RNA oligo: CAGGACUGAG AAGAAGUAAAACCdGdT, antisense RNA oligo: ACGGUUUUACUUCUUCUCAGUCCUGUA), DsiL3 (sense RNA oligo: CAGCCCUUCAAUUACCCAUAUCCdCdC, antisense RNA oligo: GGGGAUA UGGGUAAUUGAAGGGCUGCU), Dsi-13.2 (sense RNA oligo: AUCUU ACCAGUGCUGAUCAUUUA dTdA, antisense RNA oligo: UAUAAAUGAUCAGCACUGGUAAGAUUG), Dsi-13.3 (sense RNA oligo: AAAGUAUACUUCCGGGGUCAAUCdTdT, antisense RNA oligo: AAGAUUGACCCCGGAAGUAUAC UUUGG), Dsi-13.9 (sense RNA oligo: CUUCCGGGGUCAAUCUUGCAACAdAdC, antisense RNA oligo: GUUGUUGC AAGAUUGACCCCGGAAGUA), and a non-targeting DsiRNA control Dsi-NC1 (Sense:5'-CGUUAAUCGCGUAUAAUACGCGUdAdT, antisense:5'-AUACGCGUAUUAUACGCGA UUAACGAC, IDT #51-01-14-03). Predesigned siRNA SmartPools targeting the 11 downregulated genes were obtained from Dharmacon and used in *Figure 4C* and *Figure 4—figure supplement 2B and C*. Each siRNA SmartPool consisted of 4 siRNAs with On-Target*plus* modification. The following SmartPools were used: L-014208–02 (NUCKS1); L-012212–00 (CAPZA1); L-018339–00 (CCT3); L-013615–00 (FSTL1); L-011548–00 (FUBP1); L-017242–00 (GNB1); L-014597–01 (NAA50); L-020893–01 (PRELID3B); L-019719–02 (SNRPE); L-003941–00 (TFRC); L-006630–00 (HIST1H1C). On-Target*plus* non-targeting control pool (D-001810–10) was used as negative control. Transfection efficiency was assessed by transfecting cells with siGLO Red (Dharmacon) followed by FACS analysis.

HeyA8 cells (and modified cells derived from parental HeyA8 cells) were seeded at 750 cells per well on a 96-well plate one day before transfection. Cells were transfected using 0.1 µl of Lipofectamine RNAiMAX reagent per well. HCT116 cells (and modified cells derived from parental HCT116 cells) were seeded at 4000 cells per well on a 96-well plate one day before transfection. 0.2 µl of Lipofectamine RNAiMAX was used for transfection. Media was changed the day after transfection.

## Soluble CD95L protein rescue experiments

NB7 cells were seeded at 500 cells per well in a 96-well plate. Next day, cells were infected with the scrambled pLKO lentiviruses or pLKO-shL1 lentiviruses at an M.O.I. of 20 (to achieve 100% transduction efficiency under conditions omitting the puromycin selection step) in the presence of 8 µg/ml polybrene and 100 ng/ml of S2 CD95L or LzCD95L for 16 hr. Media was replaced the next day with media containing varying concentrations of recombinant CD95L.

## Real-time PCR

Total RNA was extracted and purified using QIAZOL Lysis reagent (QIAGEN) and the miRNeasy kit (QIAGEN). 200 ng of total RNA was used to generate cDNA using the high-capacity cDNA reverse Transcription kit (Applied Biosystems #4368814). cDNA was quantified using Taqman Gene expression master mix (ThermoFisher Scientific #4369016) with specific primers from ThermoFisher Scientific for GAPDH (Hs00266705_g1), human CD95 for *Figure 5—figure supplement 3* (Hs00163653_m1), human CD95 3'UTR in *Figure 2F* (custom probe, Fr primer: *GGCTAACCCCAC TCTATGAATCAAT*, Rev primer: *GGCCTGCCTGTTCAGTAACT*, Probe: *CCTTTTGCTGAAATATC*), human CD95L (Hs00181226_g1 and Hs00181225_m1), the shL3 target site in CD95L in *Figure 2D* (custom probe, Fr primer: *GGTGGCCTTGTGATCAATGAAA*, Rev primer: *GCAAGATTGACCCCG-GAAG TATA*, Probe: *CTGGGCTGTACTTTGTATATT*), and downstream of the shL3 site in *Figure 2D* (custom probe, Fr primer: *CCCCAGGATCTGGTGATGATG*, Rev primer: *ACTGCCCCCAGGTAGCT*, Probe: *CCCACATCTGCCCAGTAGT*).

To perform arrayed real-time PCR (*Figure 4—figure supplement 1*), total RNA was extracted and used to make cDNA as described for standard real-time PCR. For Taqman Low Density Array (TLDA) profiling, custom-designed 384-well TLDA cards (Applied Biosystems #43422489) were used and processed according to the manufacturer's instructions. Briefly, 50 µl cDNA from each sample (200 ng total input RNA) was combined with 50 µl TaqMan Universal PCR Master Mix (Applied

Biosystems) and hence a total volume of 100 µl of each sample was loaded into each of the 8 sample loading ports on the TLDA cards that were preloaded with assays from Thermofisher Scientific for human GAPDH control (Hs99999905_m1) and for detection of ATP13A3 (Hs00225950_m1), CAPZA1 (Hs00855355_g1), CCT3 (Hs00195623_m1), FSTL1 (Hs00907496_m1), FUPB1 (Hs00900762_m1), GNB1 (Hs00929799_m1), HISTH1C (Hs00271185_s1), NAA50 (Hs00363889_m1), NUCKS1 (Hs01068059_g1), PRELID3B (Hs00429845_m1), SNRPE (Hs01635040_s1), and TFRC (Hs00951083_m1) after the cards reached room temperature. The PCR reactions were performed using Quantstudio 7 (ThermoFisher Scientific). Since each of the port loads each sample in duplicates on the TLDA card and because two biological replicates of each sample were loaded onto two separate ports, quadruplicate Ct values were obtained for each sample. Statistical analysis was performed using Student's t test. Cells were plated at 600,000 per 15 mm dish (Greiner CELLSTAR, cat#P7237, Sigma) after one day of puromycin selection. Total RNA was harvested at 50 hrs after plating for RNAseq analysis.

## Western blot analysis

Protein extracts were collected by lysing cells with RIPA lysis buffer (1% SDS, 1% Triton X-100, 1% deoxycholic acid). Protein concentration was quantified using the DC Protein Assay kit (Bio-Rad, Hercules, CA). 30 µg of protein were resolved on 8–12% SDS-PAGE gels and transferred to nitrocellulose membranes (Protran, Whatman) overnight at 25 mA. Membranes were incubated with blocking buffer (5% non-fat milk in 0.1% TBS/Tween-20) for 1 hr at room temperature. Membranes were then incubated with the primary antibody diluted in blocking buffer over night at 4°C. Membranes were washed 3 times with 0.1% TBS/Tween-20. Secondary antibodies were diluted in blocking buffer and applied to membranes for 1 hr at room temperature. After 3 more additional washes, detection was performed using the ECL reagent (Amersham Pharmacia Biotech) and visualized with the chemiluminescence imager G:BOX Chemi XT4 (Synoptics).

## CD95 surface staining

Cell pellets of about $10^6$ cells were resuspended in about 100 µl of PBS on ice. After resuspension, 4 µl of either anti-CD95 primary antibody (BD #556640) conjugated with fluorescein isothiocyanate (FitC), or the matching Isotype control (BD #551954), Mouse IgG1 κ conjugated with FitC, were added. Cells were incubated on ice at 4°C, in the dark, for 25 min, washed twice with PBS, and percent green cells were determined by flow cytometry (Becton, Dickinson).

## Cell death quantification (DNA fragmentation)

A cell pellet (500,000 cells) was resuspended in 0.1% sodium citrate, pH 7.4, 0.05% Triton X-100, and 50 µg/ml propidium iodide. After resuspension, cells were incubated 2 to 4 hrs in the dark at 4°C. The percent of subG1 nuclei (fragmented DNA) was determined by flow cytometry.

## Cell growth and fluorescence over time

After treatment/infection, cells were seeded at 500 to 4000 per well in a 96-well plate at least in triplicate. Images were captured at indicated time points using the IncuCyte ZOOM live cell imaging system (Essen BioScience) with a 10x objective lens. Percent confluence, red object count, and the green object integrated intensity were calculated using the IncuCyte ZOOM software (version 2015A).

## RNA-Seq analysis

The following describes the culture conditions used to produce samples for RNA-Seq in *Figure 4*. HeyA8 ΔshR6 clone #11 cells were infected with pLKO-shScr or pLKO-shR6. A pool of three 293T ΔshL3 clones was infected with either pTIP-shScr or pTIP-shL3. After selection with puromycin for 2 days, the pTIP-infected 293T cells were plated with Dox in duplicate at 500,000 cells per T175 flask. The pLKO-infected HeyA8 cells were plated at 500,000 cells per flask. Total RNA was harvested 50 hrs and 100 hrs after plating. In addition, 293T cells were infected with either pLKO-shScr or pLKO-shL1 and RNA was isolated (100 hrs after plating) as described above for the infection with shR6. Finally, HeyA8 cells were transfected with RNAiMAX in 6-wells with siScr (NT2) or siL3

oligonucleotides (Dharmacon) at 25 nM. The transfection mix was removed after 9 hrs and RNA harvested 48 hrs after transfection.

Total RNA was isolated using the miRNeasy Mini Kit (Qiagen, Cat.No. 74004) following the manufacturer's instructions. An on-column digestion step using the RNAse-free DNAse Set (Qiagen, Cat. No.: 79254) was included for all RNA-Seq samples.

RNA libraries were generated and sequenced (Genomics Core facility at the University of Chicago). The quality and quantity of the RNA samples were checked using an Agilent bio-analyzer. Paired end RNA-SEQ libraries were generated using Illumina TruSEQ TotalRNA kits using the Illumina provided protocol (including a RiboZero rRNA removal step). Small RNA-SEQ libraries were generated using Illumina small RNA SEQ kits using the Illumina provided protocol. Two types of small RNA-SEQ sub-libraries were generated: one containing library fragments 140–150 bp in size and one containing library fragments 150–200 bp in size (both including the sequencing adaptor of about 130 bp). All three types of libraries (one RNA-SEQ and two small RNA-SEQ) were sequenced on an Illumina HiSEQ4000 using Illumina provided reagents and protocols. Adaptor sequences were removed from sequenced reads using TrimGalore (https://www.bioinformatics.babraham.ac.uk/projects/trim_galore), and the trimmed reads were mapped to the hg38 assembly of the human genome with Tophat and bowtie2. Raw read counts were then assigned to genes using HTSeq. Differential gene expression was analyzed with the R Bioconductor DESeq2 package (*Love et al., 2014*) using shrinkage estimation for dispersions and fold changes to improve stability and interpretability of estimates. P values and adjusted P values were calculated using the DESeq2 package.

To identify differentially abundant RNAs in cells expressing either shL3 or shR6, using a method unbiased by genome annotation, we also analyzed the raw 100 bp reads for differential abundance. First, the second end in each paired end read was reverse complemented, so that both reads were on the same strand. Reads were then sorted and counted using the core UNIX utilities sort and uniq. Reads with fewer than 128 counts across all 16 samples were discarded. A table with all of the remaining reads was then compiled, summing counts from each sequence file corresponding to the same sample. This table contained a little over 100,000 reads. The R package edgeR (http://bioinformatics.oxfordjournals.org/content/26/1/139) was used to identify differentially abundant reads, and then these reads were mapped to the human genome using blat (http://genome.cshlp.org/content/12/4/656.abstract) to determine chromosomal location whenever possible. Homer (http://homer.salk.edu/homer/) was used to annotate chromosomal locations with overlapping genomic elements (such as genes). Raw read counts in each sequence file were normalized by the total number of unique reads in the file.

To identify the most significant changes in expression of RNAs both methods of RNAs-Seq analyses (alignment and read based) were used to reach high stringency. All samples were prepared in duplicate and for each RNA the average of the two duplicates was used for further analysis. In the alignment-based analysis, only RNAs that had a base mean of >2000 reads and were significantly deregulated between the groups (adjusted p-value<0.05) were considered for further analysis. RNAs were scored as deregulated when they were more than 1.5 fold changed in the shL3 expressing cells at both time points and in the shR6 expressing cells at either time points (each compared to shScr expressing cells) (*Supplementary file 1*). This was done because we found that the pLKO-driven expression of shR6 was a lot lower than the pTIP-driven expression of shL3 (see the quantification of the two shRNAs in *Figure 5—figure supplement 1A*). This likely was a result of the reduced cellular responses in the shR6 expressing cells. In the read-based analysis, reads were only considered if they had both normalized read numbers of >10 across the samples in each treatment, as well as less than two fold variation between duplicates and >1.5 fold change between treatment groups at both time points and both cell lines (*Supplementary file 1*). After filtering, reads were mapped to the genome and associated with genes based on chromosomal localization. Finally, all RNAs were counted that showed deregulation in the same direction with both methods. This resulted in the identification of 11 RNAs that were down and 1 that was upregulated in cells exposed to the shRNAs shL3 and shR6. To determine the number of seed matches in the 3'UTR of downregulated genes, the 3'UTRs of the 11 mRNAs were extracted from the Homo sapiens gene (GRCh38.p7) dataset of the Ensembl 86 database using the Ensembl Biomart data mining tool. For each gene, only the longest deposited 3'UTR was considered. Seed matches were counted in all 3'UTRs using inhouse Perl scripts.

GSEA used in *Figure 4D* was performed using the GSEA v2.2.4 software from the Broad Institute (http://software.broadinstitute.org/gsea); 1000 permutations were used. The Sabatini gene lists (*Supplementary file 2*) were set as custom gene sets to determine enrichment of survival genes versus the nonsurvival control genes in downregulated genes from the RNA seq data; Adjusted p-values below 0.05 were considered significantly enriched. The GO enrichment analysis shown in *Figure 4F* was performed using all genes that after alignment and normalization were found to be at least 1.5 fold downregulated with an adjusted p values of < 0.05, using the software available on www.Metascape.org and default running parameters.

## Conversion of shL3 and shR6 to siRNAs

From the RNA-Seq analysis with HeyA8 ΔshR6 infected with pLKO-shR6 and 293T ΔshL3 clones infected pTIP-shL3, we analyzed the mature double-stranded RNAs derived from pLKO-shR6 and pTIP-shL3 and found that the most abundant RNA forms were both shifted by one nucleotide. Based on these most abundant species observed after cellular processing, we converted shL3 and shR6 sequences to siRNAs. The genomic target sequence in shL3 (21nt) is 5'-ACUGGGCUGUACUUUGUAUAU-3'. For the shL3 => siL3 sense strand, one G was added before the A on the 5' end while the last U on the 3' end was deleted, and second and third to the last ribonucleotides on the 3' end (UA) were replaced with deoxyribonucleotides for stabilization. For shL3 => siL3 antisense strand, the last three nucleotides on the 5' end (AUA) were deleted and one U and two dTs (UdTdT) were added after the last U on the 3'end. The shL3 => siL3 sense strand is 5'- GACUGGGCUGUACUUUGUAdTdA-3' and antisense strand is 5'-/5Phos/UACAAAGUACAGCCCAGUUdTdT-3'. The shR6 => siRNA was designed in a similar fashion except that two Gs instead of one G were added to the 5' end of the sense strand while UUdTdT instead of UdTdT was added to the 3' end of the antisense strand. The genomic target sequence in shR6 (21nt) is 5'-GUGCAGAUGUAAACCAAACUU-3'. The shR6 => siR6 sense strand is 5'-GGGUGCAGAUGUAAACCAAAdCdT-3' and antisense strand is 5'-/5Phos/UUUGGUUUACAUCUGCACUUdTdT-3'. Both shL3 => siL3 and ShR6 => siR6 siRNA duplexes were purchased from Dharmacon.

## Construction of pTIP-shRNA libraries

The pTIP-shRNA libraries were constructed by subcloning libraries of 143nt PCR inserts of the form 5'-*XXXXXXXXXXXXXXXXXXXXXXXXXXXXXXATAGAGATCGNNNNNNNNNN NNNNNNNNNNNNNCTCGAGNNNNNNNNNNNNNNNNNNNNNNNNTTTTTGTACCGAGCTCGGATCCACTAGTCCAGTGTGGGCATGCTGCGTTGACATTGATT*-3' into the pTIP-shR6 vector after excising the shR6 insert. The poly-N region represents the 21-mer sense and antisense shRNA hairpin. The intervening CTCGAG is the loop region of the shRNA. The 5 libraries targeting Venus, CD95L ORF, CD95L 3'UTR, CD95 ORF, or CD95 3'UTR were composed of every possible 21-mer shRNA (i.e. each nearest neighbor shRNA was shifted by 1 nucleotide). These libraries were synthesized together on a chip as 143 bp single-stranded DNA oligos (CustomArray Inc, Custom 12K oligo pool). Each shRNA pool had its own unique 5' end represented by the poly-*X* region. This allowed selective amplification of a particular pool using 1 of 5 unique Fr primers (CD95L ORF: 5'-*TGGCTTTATATATCTCCCTATCAGTG*-3', CD95L 3' UTR: 5'-*GGTCGTCCTATCTATTATTATTCACG*-3', CD95 ORF: 5'-*TCTTGTGTCCAGACCAATTTATTTCG*-3', CD95 3'UTR: 5'-*CTCATTGACTATCGTTTTAGCTACTG*-3', Venus: 5'-*TATCATCTTTCATGATGACTTTCCGG*-3') and the common reverse primer 5'-*AATCAATGTCAACGCAGCAT*-3'. Phusion High Fidelity Polymerase (NEB #M0530) was used to amplify each library pool; standard PCR conditions were used with an annealing temperature of 61°C and 15 cycles. PCR reactions were purified using PCR Cleanup kit (QIAGEN). The pTIP-shR6 vector and each of the amplified libraries were digested with SphI-HF and BsaBI. Digested PCR products were run on either a 2% Agarose gel or a 20% polyacrylamide (29:1) gel made with 0.5 x TBE buffer. PCR products were extracted using either Gel Extraction kit (QIAGEN) for extraction from Agarose gels or via electroelution using D-Tube Dialyzer Mini columns (Novagen #71504). Purified PCR inserts were then ligated to the linearized pTIP vector with T4 DNA ligase for 24 hr at 16°C. The ligation mixtures were transformed via electroporation in MegaX DH10B T1 cells (Invitrogen #C6400) and plated on 24 cm ampicillin dishes. At least 10 colonies per pool were picked and sequenced to verify successful library construction. After verification, all colonies per library were pooled together and plasmid

DNA extracted using the MaxiPrep kit (QIAGEN). The 5 pTIP-shRNA library DNA preps were used to produce virus in 293T cells.

## Lethality screen with pTIP-shRNA libraries

NB7 cells were seeded at $1.5 \times 10^6$ per 145 cm$^2$ dish. Two dishes were infected with each of the 5 libraries with a transduction efficiency of about 10% to 20%. Media was replaced next day. Infected cells were selected with 1.5 µg/ml puromycin. Cells infected with the Venus, CD95L ORF, and CD95L 3'UTR-targeting libraries were pooled in a 1:1:1 ratio to make the CD95L cell pool. Likewise, cells infected with the Venus, CD95 ORF, and CD95 3'UTR-targeting libraries were pooled to make the CD95 receptor cell pool. The CD95 and the CD95L cell pools were plated separately each in 2 sets of duplicates seeded at 600,000 cells per 145 cm$^2$ dish. One set received 100 ng/ml Dox, and the other one was left untreated (total of 4 dishes per combined pool; 2 received no treatment and 2 received Dox). Cells infected with the different libraries were also plated individually in triplicate with or without Dox on a 96-well plate to assess the overall toxicity of each pool. DNA was collected from each 145 cm$^2$ dish 9 days after Dox addition.

The shRNA barcodes were amplified from the harvested DNA template using NEB Phusion Polymerase with 4 different pairs of primers (referred to as N, N + 1, N + 2, and N + 3) in separate reactions per DNA sample. The N pair consisted of the primers originally used to amplify the CD95L ORF library (Fr: 5'-*TGGCTTTATATATCTCCCTATCAGTG*-3' and Rev: 5'-*AATCAATGTCAACG-CAGCAT*-3'). The N + 1 primers had a single nucleotide extension at each 5' end of the N primers corresponding to the pTIP vector sequence (Fr: 5'-*TTGGCTTTATATATCTCCCTATCAGTG*-3' and Rev: 5'-*TAATCAATGTCAACGCAGCAT*-3'). The N + 2 primers had 2 nucleotide extensions (Fr: 5'-*C TTGGCTTTATATATCTCCCTATCAGTG*-3' and Rev: 5'-*ATAATCAATGTCAACGCAGCAT*-3'), and the N + 3 primers had 3 nucleotide extensions (Fr: 5'-*TCTTGGCTTTATATATCTCCCTATCAGTG*-3' and Rev: 5'-*AATAATCAATGTCAACGCAGCAT*-3'). The barcodes from the pTIP-shRNA library plasmid preparations were also amplified using Phusion Polymerase with the N, N + 1, N + 2, and N + 3 primer pairs. The shRNA barcode PCR products were purified from a 2% Agarose gel and submitted for 100 bp paired-end deep sequencing (Genomics Core facility at the University of Chicago). DNA was quantitated using the Qubit. The 4 separate PCR products amplified using N, N + 1, N + 2, and N + 3 were combined in equimolar amounts for each sample. Libraries were generated using the Illumina TruSeq PCR-free kit using the Illumina provided protocol. The libraries were sequenced using the HiSEQ4000 with Illumina provided reagents and protocols. Raw sequence counts for DNAs were calculated by HTSeq. shRNA sequences in the PCR pieces of genomic DNA were identified by searching all reads for the sense sequence of the mature shRNA plus the loop sequence CTCGAG. To avoid a division by zero problem during the subsequent analyses all counts of zero in the raw data were replaced with 1. A few sequences with a total read number <10 across all plasmids reads were not further considered. In the CD95L pool this was only one shRNA (out of 2362 shRNAs) (L792') and in the CD95 20 shRNAs (out of 3004 shRNAs) were not represented (R88, R295, R493, R494, R496, R497, R498, R499, R213', R215', R216', R217', R220', R221', R222', R223', R225', R226', R258', R946', R1197', R423'). While most shRNAs in both pools had a unique sequence two sequences occurred 6 times (L605', L607', L609', L611', L613', L615', and L604', L606', L608', L610', L612', L614'). In these cases, read counts were divided by 6. Two shRNAs could not be evaluated: 1) shR6 in the CD95 pool. It had a significant background due to the fact that pTIP-shR6 was used as a starting point to clone all other shRNAs. 2) shL3 was found to be a minor but significant contaminant during the infection of some of the samples. For each condition, two technical duplicates and two biological duplicates were available. To normalize reads to determine the change in relative representation of shRNAs between conditions, the counts of each shRNA in a subpool (all replicates and all conditions) was divided by the total number of shRNAs in each subpool (%). First, the mean of the technical replicates (R1 and R2) was taken. To analyze the biological replicates and to determine the changes between conditions, two analyses were performed: 1) The change in shRNA representation between the cloned plasmid library and cells infected with the library and then cultured for 9 days without Dox (infection -Dox). Fold downregulation was calculated for each subpool as [(plasmid %/-Dox1 %+plasmid %/-Dox2 %)/2]. 2) The difference in shRNA composition between the infected cells cultured with (infection +Dox) and without Dox. Fold downregulation was calculated for each subpool as [(-Dox1 %/+Dox1%)+(-Dox1 %/+Dox2%)+(-Dox2 %/+Dox1%)+(-Dox2 %/+Dox2%)/4].

Only shRNAs were considered that were at least 5-fold underrepresented in either of the two analyses (data in *Supplementary file 3*).

## The toxicity index (TI) and GC content analysis

The TI in *Figure 7A* is defined by the sum of the counts of a 6mer or 8mer seed match in the 3'UTRs of critical survival genes divided by the seed match counts in the 3'UTRs of nonsurvival genes. We used the 1882 survival genes recently described in a CRISPR/Cas9 lethality screen by Wang et al. (*Wang et al., 2015*). The survival genes were defined by having a CRISPR score of $<-0.1$ and an adjusted p-value of $<0.05$. We chose as a control group to these top essential genes the bottom essential genes using inverse criteria (CRISPR score of $>0.1$ and adjusted p-value of $<0.05$) and are referring to them as the 'nonsurvival genes'. Both counts were normalized for the numbers of genes in each gene set. 3'UTRs were retrieved as described above. For the survival genes 1846 and for the nonsurvival genes 416 3'UTRs were found. For each gene, only seed matches in the longest 3'UTR were counted. The TI was calculated for each of the 4096 possible 6mer combinations and each of the 65536 possible 8mer combinations (*Supplementary file 4*). These numbers were then assigned to the results of the shRNA screen (*Supplementary file 5*). An alternative TI was calculated in *Figure 7—figure supplement 1B* and is based on the top 850 most highly expressed survival genes (all expressed >1000 average reads) and 850 expression matched genes not described to be critical for cancer cell survival were selected as controls.

For the analyses in *Figure 7C and D*, the GC content % was calculated for every 6mer in the CD95L ORF shRNA pool. The GC content % was then plotted against the log(Fold down) for each shRNA in the CD95L ORF shRNA after infection (compared to the plasmid composition) in *Figure 7C* and after addition of Dox (compared to cells infected but not treated with Dox) in *Figure 7D*. In *Figure 7E*, the log(TI) and GC content % was extracted for every possible 6mer and plotted. Pearson correlation coefficient and associated p-value were calculated in R3.3.1.

## Sylamer analysis

Sylamer is a tool to test for the presence of RNAi-type regulation effects from a list of differentially expressed genes, independently from small RNA measurements (*van Dongen et al., 2008*) (http://www.ebi.ac.uk/research/enright/software/sylamer). For short stretches of RNA (in this case length 6, 7, and 8 in length corresponding to the lengths of the determinants of seed region binding in RNAi-type binding events), Sylamer tests for all possible motifs of this length whether the motif occurrences are shifted in sequences associated with the list under consideration, typically 3'UTRs when analyzing RNAi-type binding events. A shift or enrichment of such a motif towards the down-regulated end of the gene list is consistent with upregulation of a small RNA that has the motif as the seed region. Sylamer tests in small increments along the list of genes, using a hypergeometric test on the counts of a given word, comparing the leading part of the gene list to the universe of all genes in the list. For full details, refer to (*van Dongen et al., 2008*). Enriched motifs stand out from the back-ground of all motifs tested, as visible in the Sylamer plot. The plot consist of many different lines, each line representing the outcomes of a series of tests for a single word, performed along regularly spaced intervals (increments of 200 genes) of the gene list. Each test yields the log-transformed P-value arising from a hypergeometric test as indicated above. If the word is enriched in the leading interval, the log-transformed value has its value plotted on the positive y-axis (sign changed), if the word is depleted the log-transformed value is plotted on the negative y-axis. 3'UTRs were used from Ensembl, version 76. As required by Sylamer, they were cleaned of low-complexity sequences and repetitive fragments using respectively Dust (*Morgulis et al., 2006*) with default parameters and the RSAT interface (*Medina-Rivera et al., 2015*) to the Vmatch program, also run with default parameters. Sylamer (version 12–342) was run with the Markov correction parameter set to 4.

## Statistical analyses

Continuous data were summarized as means and standard deviations (except for all IncuCyte experiments where standard errors are shown) and dichotomous data as proportions. Continuous data were compared using t-tests for two independent groups and one-way ANOVA for 3 or more groups. For evaluation of continuous outcomes over time, two-way ANOVA was used with one

factor for the treatment conditions of primary interest and a second factor for time treated as a categorical variable to allow for non-linearity. Comparisons of single proportions to hypothesized null values were evaluated using binomial tests. Statistical tests of two independent proportions were used to compare dichotomous observations across groups.

The effects of treatment on wild-type versus either Dicer$^{-/-}$ or Drosha$^{-/-}$ cells were statistically assessed by fitting regression models that included linear and quadratic terms for value over time, main effects for treatment and cell type, and two- and three-way interactions for treatment, cell-type and time. The three-way interaction on the polynomial terms with treatment and cell type was evaluated for statistical significance since this represents the difference in treatment effects over the course of the experiment for the varying cell types.

To test if higher TI is enriched in shRNAs that were highly downregulated, p-values were calculated based on permutated datasets using Mann-Whitney U tests. The ranking of TI was randomly shuffled 10,000 times and the W statistic from our dataset was compared to the distribution of the W statistic of the permuted datasets. Test of enrichment was based on the filtered data of at least 5-fold difference, which we define as a biologically meaningful. Fisher Exact Tests were performed to assess enrichment of downregulated genes (i.e. >1.5 downregulated with adjusted p-value<0.05) amongst genes with at least one si/shRNA seed match. All statistical analyses were conducted in Stata 14 (RRID:SCR_012763) or R 3.3.1 in Rstudio (RRID:SCR_000432).

## Acknowledgements

We are grateful to Lindsay Stolzenburg and Ann Harris for helping to set up the CRISPR/Cas9 gene editing method and to Matthew Schipma for computational support. We would like to thank the Gene Editing and Screening Core, at Memorial Sloan Kettering in New York City, NY, for RNAi reagents and services. MH was supported by the Intramural Research Program of NIAMS. AAS. acknowledges support by the Swedish Research Council postdoctoral fellowship. This work was funded by training grants T32CA070085 (to MP) and T32CA009560 (to WP) R50CA211271 (to JCZ), and R35CA197450 (to MEP).

## Additional information

### Funding

| Funder | Grant reference number | Author |
| --- | --- | --- |
| National Institutes of Health | R35CA197450 | Marcus E Peter |
| Svenska Forskningsrådet Formas | | Aishe A Sarshad |
| National Institutes of Health | T32CA009560 | William Putzbach |
| National Institutes of Health | T32CA070085 | Monal Patel |
| National Institutes of Health | R50CA211271 | Jonathan C Zhao |

The funders had no role in study design, data collection and interpretation, or the decision to submit the work for publication.

### Author contributions

William Putzbach, Formal analysis, Investigation, Writing—review and editing, Designed experiments, Carried out experiments and collected the data, Assisted with preparing the manuscript; Quan Q Gao, Formal analysis, Investigation, Writing—review and editing, Designed experiments, Carried out experiments and collected the data, Assisted with preparing the manuscript Contributed equally with: William Putzbach; Monal Patel, Investigation, Writing—review and editing, Designed experiments, Carried out experiments and collected the data, Assisted with preparing the manuscript, Established the arrayed qPCR method and performed experiments Contributed equally with: William Putzbach; Stijn van Dongen, Data curation, Formal analysis, Validation, Performed the SYMALER analyses; Ashley Haluck-Kangas, Formal analysis, Investigation, Carried out data analysis; Aishe A Sarshad, Formal analysis, Investigation, Performed AGO pull down experiments; Elizabeth T

Bartom, Data curation, Formal analysis, Validation, Investigation, Assisted with design of computational analyses, Performed data base and RNA-Seq analysis and computational analyses; Kwang-Youn A Kim, Data curation, Formal analysis, Provided biostatistics support, Determined statistical significance; Denise M Scholtens, Formal analysis, Provided biostatistics support, Determined statistical significance, performed GSEA; Markus Hafner, Resources, Investigation, Provided assistance and discussions on the mechanism of RNAi and the RISC; Jonathan C Zhao, Conceptualization, Data curation, Formal analysis, Supervision, Writing—original draft, Project administration, Writing—review and editing, Performed data base and RNA-Seq analysis; Andrea E Murmann, Investigation, Writing—review and editing, Carried out experiments and collected the data, Assisted with preparing the manuscript; Marcus E Peter, Conceptualization, Data curation, Formal analysis, Supervision, Investigation, Writing—original draft, Project administration, Writing—review and editing, Wrote the paper, Designed all experiments, Supervised the project

### Author ORCIDs
William Putzbach, http://orcid.org/0000-0002-2669-6654
Quan Q Gao, http://orcid.org/0000-0002-3316-2968
Monal Patel, http://orcid.org/0000-0003-1525-9915
Kwang-Youn A Kim, https://orcid.org/0000-0002-4168-757X
Markus Hafner, https://orcid.org/0000-0002-4336-6518
Marcus E Peter, http://orcid.org/0000-0003-3216-036X

### Decision letter and Author response
Decision letter https://doi.org/10.7554/eLife.29702.032
Author response https://doi.org/10.7554/eLife.29702.033

# Additional files

### Supplementary files
• Supplementary file 1. Results of the RNA-Seq analysis used to generate *Figure 4B*.
DOI: https://doi.org/10.7554/eLife.29702.022

• Supplementary file 2. Gene lists used in this work.
DOI: https://doi.org/10.7554/eLife.29702.023

• Supplementary file 3. shRNA screen data.
DOI: https://doi.org/10.7554/eLife.29702.024

• Supplementary file 4. The 6mer and 8mer toxicity index.
DOI: https://doi.org/10.7554/eLife.29702.025

• Supplementary file 5. Correlation between experimental shRNA toxicity and TI.
DOI: https://doi.org/10.7554/eLife.29702.026

• Transparent reporting form
DOI: https://doi.org/10.7554/eLife.29702.027

### Major datasets
The following dataset was generated:

| Author(s) | Year | Dataset title | Dataset URL | Database, license, and accessibility information |
| --- | --- | --- | --- | --- |
| Peter M | 2017 | Cell Death-Inducing Sequences with RNAi Activity Embedded in the Genes of CD95 and CD95L | https://www.ncbi.nlm.nih.gov/geo/query/acc.cgi?acc=GSE87817 | Publicly available at NCBI Gene Expression Omnibus (accession no: GSE87817) |

The following previously published datasets were used:

| Author(s) | Year | Dataset title | Dataset URL | Database, license, and accessibility information |
|---|---|---|---|---|
| Cowley GS, Weir BA, Vazquez F, Tamayo P, Scott JA, Rusin S, East-Seletsky A, Ali LD, Gerath WFJ, Pantel SE, Lizotte PH, Jiang G, Hsiao J, Tsherniak A, Dwinell E, Aoyama S, Okamoto M, Harrington W, Gelfand E, Green TM, Tomko MJ, Gopal S, Wong TC, Li H, Howell S, Stransky N, Liefeld T, Jang D, Bistline J, Meyers BH, Armstrong SA, Anderson KC, Stegmaier K, Reich M, Pellman D, Boehm JS, Mesirov JP, Golub TR, Root DE, Hahn WC | 2014 | Achilles_QC_v2.4.3.rnai.shRNA. table.txt (2) Achilles_QC_v2.4.3.rnai. gct | http://dx.doi.org/10. 6084/m9.figshare. 1019859 | Available at figshare under a CC0 Public Domain licence (https://figshare.com/ ) |

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
