## [Decision Letter]

Thank you for submitting your article "CD95L derived si- and shRNAs and the CD95L mRNA kill cancer cells through an RNAi mechanism by targeting survival genes" for consideration by *eLife*. Your article has been reviewed by three peer reviewers, and the evaluation has been overseen by a Reviewing Editor and Charles Sawyers as the Senior Editor. The following individuals involved in review of your submission have agreed to reveal their identity: John G Doench (Reviewer #1); Gregory J Hannon (Reviewer #3).

The reviewers have discussed the reviews with one another and the Reviewing Editor has drafted this decision to help you prepare a revised submission.

Summary:

Previously, the authors conducted a study that described DICE ("Death induced by CD95R/L elimination"), in which they show that depletion of these genes by RNAi results in cell death across a dozen cancer cell lines (Hadji et al., 2014). Here, they follow-up on this observation by fine-mapping the cause of DICE and show that the viability effect triggered by these RNAi reagents is actually not dependent on the target genes, but rather on a seed-based (i.e. microRNA-like) off-target effect of the RNAi reagents, leading to a new acronym, DISE (death induced by survival gene elimination).

Essential revisions:

1) The consensus view among reviewers is that the emphasis in the manuscript is misplaced. It is largely about off-target effects of RNAi reagents, and has little to do with CD95 or CD95L. If the conclusion of this manuscript is that some seeds are more likely to have viability effects than others, then the focus should be on that conclusion. As written, the manuscript is very CD95-centric, which implies that there is something special about this gene.

2) Since this manuscript is really about generic off-target effects, then there are additional resources that could be used to better analyze this phenomenon. For example, the TRC library has been screened against hundreds of cell lines and those data are available (Cowley et al., Sci. Data, 2014), and it would be important to thoroughly analyze those data to see if these observations here generalize well.

3) Additionally, as mentioned in the last paragraph of the subsection “DISE is caused by loading of the guide strand of toxic si/shRNAs into the RISC”, the inactivity of miR-30 backbone shRNAs to produce this response deserves much more treatment than a "data not shown." If the off-target effects arise because of something specific to the TRC shRNAs used, that mechanism needs to be characterized. If it is purely an expression-level difference, that needs to be documented. One potential experimental approach could use transient transfection of siRNAs – which can be used at different doses, or with pools of siRNAs to represent the different Dicer products of TRC shRNAs.

4) The authors produced 3 different target deletions (ΔsiL3, ΔshL3 and ΔshR6) in 293T or HeyA8 cells to show that expression of the si or shRNAs, even in the absence of the target, is still lethal. The deletions seem to produce frame-shift mutations in the coding sequence (as the nt deleted are not a multiple of 3). The authors indeed show that ΔshR6 produces a protein knockout, which they use for a different experiment (Figure 1—figure supplement 3). This provokes the question of whether this is also the case for ΔsiL3 and ΔshL3 and whether it is appropriate to call them "target site" mutants if they are indeed functional knockouts. The authors state in their Introduction "after deleting the CD95 gene tumors barely grew in vivo (Chen et al., 2010; Hadji et al., 2014)" so maybe they could comment on the fact that HeyA8 ΔshR6, being a functional knockout and not exclusively a target site mutant grows similar to the parental cell line (Figure 1).

5) In Figure 6 enforced expression of the CD95L mRNA is toxic in HeyA8 cells. However, in Figure 1—figure supplement 2, the CD95L-WT ORF is expressed (together with a scramble shRNA) and there is no observable increase in% subG1 compared to expression of vector only (together with a scramble shRNA) in either NB7 cells (panel D) or MCF-7 (panel G). Could the authors please comment on whether there are any other variables we should take into account or whether this is a cell type-specific phenotype. It is difficult to understand that NB7 lack caspase-8 but if they show the toxicity phenotype upon expression of the si/shRNAs and if over-expression of the CD95L is mediating the same phenotype, the model would predict CD95L would still cause lethality of this cell line.

6) The authors perform a pooled screen to identify the most toxic sequences in the CD95 and CD95L genes (Figure 5). The pool screen analysis is performed comparing the final time points to the plasmid pool. This is concerning as virus production and infection could also lead to biases in the shRNA representation. As it stands, it could very well be that many of the "depleted" shRNAs were never present in the cells to begin with. This leads to particular concern about the results in the absence of doxycycline. The authors state that the depletion observed without adding dox is most likely due to leakiness in the Tet-on system (subsection “Identification of toxic shRNAs in the CD95L and CD95 mRNAs”, second paragraph). However, in their initial experiments (Figure 1) they expressed the highly toxic shRNA shL3 in 293T without any evidence of toxicity for at least 4 days. Thus, it would be very important to show the depletions observed in this pooled screen are indeed biological. Since the authors do not seem to have an initial post-infection time point to compare to, they could produce virus following the same set up and show that plasmid representation roughly corresponds to the shRNAs cloned from cells right after infection.

7) Regarding the analysis of AGO-bound small RNAs that align to the CD95L, does the analysis take into account the possibility of PCR duplicates? Also, strand information is essential if the model is that this CD95L derived small RNAs are mediating knockdown of survival genes. The analysis text does not specify whether BLAST was run with any strand specificity so it would be essential for the authors to clarify.

---

## [Author Response]

Essential revisions:1) The consensus view among reviewers is that the emphasis in the manuscript is misplaced. It is largely about off-target effects of RNAi reagents, and has little to do with CD95 or CD95L. If the conclusion of this manuscript is that some seeds are more likely to have viability effects than others, then the focus should be on that conclusion. As written, the manuscript is very CD95-centric, which implies that there is something special about this gene.

We agree with the reviewers that what started out as a CD95 project has turned into an RNAi toxicity project. However, the starting point was indeed CD95 centric, as it developed from the observation that >80% of all tested si/shRNAs designed to target either CD95 or CD95L were toxic to all cancer cell lines, even in the absence of the targets. Based on these data we were faced with the question: in general are 80% of all commercially designed si/shRNAs toxic to cells via an off-target effect or is there something special about CD95 and CD95L? It was unlikely that the majority of all si/shRNAs targeting a gene would be toxic. If that were the case, RNAi would not have become such a widely-used tool. As this manuscript contains mostly data on si/shRNAs derived from CD95 or CD95L, we must discuss aspects of this work in the context of CD95/CD95L. However, we have now modified the manuscript to make clear that CD95 and CD95L serve as a model to study the toxic effects of si/shRNAs containing certain seed sequences. We have restructured the Introduction and whenever possible, in the context of a section, we are now referring to toxic seed containing si/shRNAs rather than to CD95/CD95L derived si/shRNAs. In fact, in a review that will be published shortly (title: "DISE – An RNAi Off-Target Effect that Kills Cancer Cells") that was made available to the reviewers, we do discuss the DISE effect in more general terms not directly related to CD95. Most importantly, following the consensus advice of the reviewers, we have changed the title of the manuscript to reflect the scope of our work that goes beyond CD95 and CD95L. Finally, we have moved data on the toxic activity of the CD95L ORF into a future manuscript to maintain the focus on toxic si/shRNAs in this one.

We agree with the reviewers in that CD95 or CD95L are not unique. We assume there must be many other coding and also noncoding genes that can contain sequences that when expressed as si/shRNAs will be toxic.

In another recent manuscript, we explored the question of whether other genes contain such toxic sequences (Patel and Peter, 2017, Cell Cycle, in press). We focused on a set of tumor suppressors (since they were least likely to be cancer survival genes) among a list of 651 survival genes that came out of our genome-wide shRNA survival screen (Hadji et al., 2014, Cell Rep. 10:208). By focusing strictly on morphological and biochemical changes typical for cells undergoing DISE as described by us before (Hadji et al., 2014, Cell Rep. 10:208), we identified five other genes (all putative tumor suppressors) for which at least two of the commercially available shRNAs killed multiple cancer cells in a way consistent with DISE. We also demonstrate in Patel and Peter that this is not a general phenomenon associated with use of TRC library shRNAs. None of the 9 available shRNAs targeting the tumor suppressors p53 or PTEN were toxic (Patel and Peter, 2017, Cell Cycle, in press). In summary, we have clarified in the revised manuscript that what we report for CD95/CD95L is likely not unique to these two genes, and that it is more likely that multiple genes in the genome contain sequences that can give rise to toxic seeds. As a result, we have refocused the manuscript away from CD95/CD95L as much as possible and now place more focus on the activities of toxic seeds.

2) Since this manuscript is really about generic off-target effects, then there are additional resources that could be used to better analyze this phenomenon. For example, the TRC library has been screened against hundreds of cell lines and those data are available (Cowley et al., Sci. Data, 2014), and it would be important to thoroughly analyze those data to see if these observations here generalize well.

We would like to thank the reviewers for directing us to this published work. We have carefully examined the cited analysis and identified the data associated with shRNAs from their screen that we also tested. We have added the result of this analysis as new Figure 1—figure supplement 1. The shRNAs we have tested here were found in this study to kill the following percentage of the 216 tested cells in their screen: shL4 (99.5%), shL1 (96.8%), shR6 (88.9%), shR7 (75%), shL2 (67.1%), shR5 (38.4%, shL5 (26.4%), and shR8 (21.3%). In addition to the shRNAs tested by us, these authors also tested an additional CD95 targeting sequence (we now call shR3). It was found to be toxic to 71.8% of the 216 cell lines. Their screen did not include shL3 but consistent with our data, shL1 and shR6 were found to be two of the most toxic shRNAs. Interestingly, the authors did not consider the data on any of the CD95L targeting shRNAs and half of the ones targeting CD95 to be significant as they received a low consistency score. A high consistency score predicts that the observed phenotype (cell death or growth reduction in this case) is caused by knocking down the targeted gene (on-target activity) (Shao et al., 2013, Genome Res, 23:665). Assuming each shRNA only targets one gene, that approach made sense. However, in our case, we know it is only the seed region and the targeted transcriptome that determines toxicity, resulting in an off-target effect. Thus, their exclusion of CD95L shRNAs only supports our conclusions.

Ours and their analysis agree that the majority of tested shRNAs targeting either CD95 or CD95L (67%) were toxic to the majority of all tested solid and hematopoietic cancer cell lines most likely through the offtarget effect we describe. The reason we identified a somewhat higher percentage of toxic shRNAs targeting CD95 and CD95L might be due to the fact that the shRNAs in our screen and in our experiments were tested individually, whereas the Cowley study used pooled libraries.

Also in agreement with our data, overall, CD95L-derived shRNAs are more toxic than CD95-derived shRNAs. Clearly the CD95L targeting shRNAs depleted in the largest percentage of the cell lines in their screen among the 9 we also tested are shL1 and shL4 (see new Figure 1—figure supplement 1). Consistent with this higher underrepresentation of CD95L-derived shRNAs, our data on the shRNA screen which identified the highest number of underrepresented and toxic shRNAs in the ORF of CD95L. We have added this result and discussion to the revised manuscript.

Finally, we would like to reiterate that the toxicity we describe is not limited to shRNAs of the TRC library. We observed similar toxicities using independent sets of siRNAs, Dsi-RNAs and also by inducible expression of shRNAs (see Figure 1—figure supplement 1, (Hadji et al., 2014, Cell Rep. 10:208)). We even successfully generated toxic siRNAs based on the processing (determined by RNA-Seq) of two of our most toxic shRNAs (shL3 and shR6) (see Figure 5—figure supplement 1). These data support the conclusion that it is not the platform or the delivery that determines toxicity but, as recognized by the reviewers, the seed sequence.

3) Additionally, as mentioned in the last paragraph of the subsection “DISE is caused by loading of the guide strand of toxic si/shRNAs into the RISC”, the inactivity of miR-30 backbone shRNAs to produce this response deserves much more treatment than a "data not shown." If the off-target effects arise because of something specific to the TRC shRNAs used, that mechanism needs to be characterized. If it is purely an expression-level difference, that needs to be documented. One potential experimental approach could use transient transfection of siRNAs – which can be used at different doses, or with pools of siRNAs to represent the different Dicer products of TRC shRNAs.

To support our statement on the activity of miR-30 based shRNAs, we have added the results of the miR-30 based shRNAs to the revised manuscript as new Figure 1—figure supplement 2. Our most toxic TRC based shRNA sequences (shL1, shL3, shL4, shR5, shR6, and shR7) were converted into miR-30 based hairpins and cloned into the Tet-inducible LT3REPIR vector by the Memorial Sloan Kettering Gene Editing and Screening Core (Dow et al., 2012, Nature Prot. 2:374). In contrast to the TRC equivalents, none of these 6 viruses affected cell growth of HeyA8 cells in a major way (new Figure 1—figure supplement 2). The conclusion that the expression of these shRNAs was too low to be toxic was based on our findings that expression of these shRNAs did not even result in a major silencing of the respective venus sensors (linked to either CD95 or the CD95L mRNA) (new Figure 1—figure supplement 2). In contrast, the toxic shRNAs shL3 and shR6 expressed from our inducible vector pTIP shut down Venus expression completely (see Figure 1—figure supplement 2).

There certainly could be other reasons for the lack in shRNA activity from these vectors outside our control. We have revised the section in the Discussion. In general, we are not concerned about the discrepancy between our data and other screens using miR-30-based shRNAs given that not only shRNAs but also siRNAs, DsiRNAs, and shRNA-based siRNAs are all effective DISE inducers. Thus, the DISE inducing activity seems to be independent of the delivery vehicle.

In response to the second part of the reviewers’ comment, we did titer siL3 to very low concentrations and the DISE inducing activity is indeed dose dependent (see new Figure 1—figure supplement 2). siL3 transfected at 0.1 nM was no longer toxic. In another manuscript in preparation we determined the IC50 for siL3 to kill cells to be 0.8 nM (Murmann et al. in preparation).

4) The authors produced 3 different target deletions (ΔsiL3, ΔshL3 and ΔshR6) in 293T or HeyA8 cells to show that expression of the si or shRNAs, even in the absence of the target, is still lethal. The deletions seem to produce frame-shift mutations in the coding sequence (as the nt deleted are not a multiple of 3). The authors indeed show that ΔshR6 produces a protein knockout, which they use for a different experiment (Figure 1—figure supplement 3). This provokes the question of whether this is also the case for ΔsiL3 and ΔshL3 and whether it is appropriate to call them "target site" mutants if they are indeed functional knockouts. The authors state in their Introduction "after deleting the CD95 gene tumors barely grew in vivo (Chen et al., 2010; Hadji et al., 2014)" so maybe they could comment on the fact that HeyA8 ΔshR6, being a functional knockout and not exclusively a target site mutant grows similar to the parental cell line (Figure 1).

All deletions were purposely designed to both delete the targeted site and to produce frame-shift mutations. In the chosen cell lines, CD95L protein and mRNA are virtually undetectable to begin with. So demonstrating functional knock-outs is only feasible for the deletion of CD95. Because the goal was to eliminate the target sites of our si/shRNAs (which we accomplished), whether the cells are true functional knock-outs (meaning, no functional protein being produced) was secondary to the question that needed to be addressed. Having functional knock-outs was an added benefit.

The reviewers are correct. Based on our in vivo mouse model data we expected cancer cell lines to not grow well after deletion of CD95. When generating the single cell clones with the target site deletion (or the complete deletion of CD95, now in future manuscript) we consistently observed that cells indeed initially grew slower than wt cells but because the effect was reduced with prolonged culture and because this effect could have been caused by clonal differences, we decided to not include this observation in the manuscript. This requires further work.

Overall, we believe we are looking at two different phenomena. Firstly, sequences derived from CD95L for instance are toxic to cancer cells regardless of the expression of the target. Secondly, as pointed out by the reviewers, we have shown in different genetically engineered mouse tumor models that the CD95 gene/protein is required for cancer cells to form tumors in vivo. We and others have shown that CD95 is involved in the maintenance of cancer stem cells (Ceppi et al., 2014, Nature Com. 5:5238; Drachsler et al., 2016, Cell Death Dis, 7:e2209; Teodoczyk et al., 2015, Cell Death Differ, 22:1192). We recently demonstrated that this occurs by activating a type I interferon/STAT1 pathway (Qadir et al., 2017, Cell Rep, 10:2373). In addition, CD95L does promote cell growth. This is apparent from the experiment in which we added either soluble CD95L (S2) (new Figure 1, left panel) or highly active leucine zipper tagged CD95L to the cells (new Figure 1, center panel). In both cases, there was a small but reproducible and dose dependent increase in growth. We are currently testing whether these activities of CD95 are required for cancer cells to grow in syngeneic immune competent mouse models, which would explain the previous observations in the genetically engineered mouse tumor models.

5) In Figure 6 enforced expression of the CD95L mRNA is toxic in HeyA8 cells. However, in Figure 1—figure supplement 2, the CD95L-WT ORF is expressed (together with a scramble shRNA) and there is no observable increase in% subG1 compared to expression of vector only (together with a scramble shRNA) in either NB7 cells (panel D) or MCF-7 (panel G). Could the authors please comment on whether there are any other variables we should take into account or whether this is a cell type-specific phenotype. It is difficult to understand that NB7 lack caspase-8 but if they show the toxicity phenotype upon expression of the si/shRNAs and if over-expression of the CD95L is mediating the same phenotype, the model would predict CD95L would still cause lethality of this cell line.

The reviewers are correct. These data seem to contradict our finding on the toxicity of the CD95L ORF. DISE kills all cancer cells and caspase-8 deficiency has no effect on this form of cell death. The reason that the cells did not seem to die in this rescue experiment is the way it was performed. In the experiments shown in former Figure 7 acute effects of CD95L ORF were monitored. Two days after infection with CD95L lentivirus, infected cells were selected for by treatment. After 24 hours in puromycin, cells were immediately plated for experiments. In contrast, in the experiment shown in Figure 1, NB7 cells were infected with the CD95L lentivirus, selected for two days with puromycin and then cultured in the presence of puromycin for another 7 days. We made two observations: First, NB7 cells seem to be less susceptible to the cell death induced by CD95L ORF and second, after 9 days in culture enough cells had recovered and expressed the siRNA resistant protein to perform the experiment. Interestingly, these CD95L expressing cells were still sensitive to DISE induction by either shL1 or shL3. We have added a sentence to the manuscript to properly describe this observation. The data on expressing CD95 in MCF-7 cells formerly in Figure 1—figure supplement 2 are not affected by this discussion as CD95 expression was not found to be toxic to any cells.

6) The authors perform a pooled screen to identify the most toxic sequences in the CD95 and CD95L genes (Figure 5). The pool screen analysis is performed comparing the final time points to the plasmid pool. This is concerning as virus production and infection could also lead to biases in the shRNA representation. As it stands, it could very well be that many of the "depleted" shRNAs were never present in the cells to begin with. This leads to particular concern about the results in the absence of doxycycline. The authors state that the depletion observed without adding dox is most likely due to leakiness in the Tet-on system (subsection “Identification of toxic shRNAs in the CD95L and CD95 mRNAs”, second paragraph). However, in their initial experiments (Figure 1) they expressed the highly toxic shRNA shL3 in 293T without any evidence of toxicity for at least 4 days. Thus, it would be very important to show the depletions observed in this pooled screen are indeed biological. Since the authors do not seem to have an initial post-infection time point to compare to, they could produce virus following the same set up and show that plasmid representation roughly corresponds to the shRNAs cloned from cells right after infection.

We discussed the design of this experiment in length during the planning of the shRNA screen and did many pilot experiments. First of all, the leakiness is not observed in all cells and not for all shRNAs, for reasons that are unclear. We found most effects of leakiness of the Tet inducible vector in HeyA8 and NB7 cells. 293T cells tolerate the vector expressing shL3 quite well and in these cells we do not see any effect on growth by just infecting them with the pTIP-shL3 vector in the absence of Dox, as seen in new Figure 2. This cell specific sensitivity to vector leakiness is the reason we documented the leakiness in NB7 cells (Figure 6—figure supplement 2) as this was most relevant to the actual screen which was performed in NB7 cells. As information for the reviewers, we chose NB7 cells for the screen because we had established the CD95L ORF venus sensor in this cell line, which allowed us to confirm shRNA knockdown activity (see Figure 6—figure supplement 1). We found the CD95L Venus sensor vector to express very small amounts of active CD95L killing CD95 apoptosis sensitive cells. To avoid any interference by this activity, we chose the caspase-8 deficient NB7 cells for the screen.

We agree with the reviewers that it would have been preferable to have a true infection control. However, due to the high variance in toxicity caused by different degrees of leakage by different shRNAs we realized that such a control was not possible. Certain shRNAs would have already been underrepresented 48 hours after infection (the time we determined that was required in NB7 cells to complete infection and virus integration). We therefore decided to compare the three conditions: 1. Plasmid pool (as a control of sequence representation in the library). 2. Cells 12 days after infection and no Dox addition (this served as the infection control the reviewers asked for, however only for shRNAs that do not show leakiness) and 3. Cells 12 days after infection in the presence of Dox (this would reveal the underrepresented/toxic shRNAs that were highly induced by Dox, both for shRNAs with and without leakiness). So, while we share the concern raised by the reviewers, we would like to point out that the data plus Dox further depleted many shRNAs when compared to the cells minus Dox. Minus Dox in this case serves as an expression control. As a side remark, we noticed that in the comprehensive cell line analysis recommended to us by the reviewers above (Cowley et al., Sci. Data, 2014) the authors also did not include an expression control but – similar to what we did – they compared the abundance of the shRNAs to the initial plasmid pool. We assume this was done for the same reason, that one cannot be certain that even at early times after infection highly active shRNAs already are not being selected against. A true infection control is therefore not possible.

The individual expression levels of the shRNAs are given in Supplementary file 3. While a very small number of shRNAs were not well expressed, the vast majority were and comparing the plus Dox with the minus Dox data we do have a relative presentation of each shRNA that takes into account infectious yield of many shRNAs. Importantly, underrepresented shRNAs identified in the comparison between library and no Dox treatment and between minus and plus Dox treatment mapped to similar regions in both CD95 and CD95L (compare Figure 6 blue with orange ticks) suggesting that the depletion is biological.

7) Regarding the analysis of AGO-bound small RNAs that align to the CD95L, does the analysis take into account the possibility of PCR duplicates? Also, strand information is essential if the model is that this CD95L derived small RNAs are mediating knockdown of survival genes. The analysis text does not specify whether BLAST was run with any strand specificity so it would be essential for the authors to clarify.

We have moved this analysis to a manuscript to be published in the future. However, we would still like to answer the questions raised by the reviewers as we take them seriously. In addition, this will also explain why we felt that all these new data would be outside the focus recommended to us by the reviewers, and they would have further increased the size of this manuscript.

While PCR duplicates are always a concern, during the analysis we made sure that the PCR amplification was in the exponential phase. We actually performed extensive pilot PCRs. When done properly, PCR duplicates are of little concern and the biases in the library are stemming mainly from the 3’ and 5’ adapter ligation steps. Unfortunately, those cannot really be avoided entirely. We have previously published a careful study on biases in small RNA library preparations (Hafner et al., 2011, RNA, 17:1697).

The analysis was done in a strand specific fashion. All reads aligned are derived from the plus strand. Very few antisense reads were detected. We are aware that this raises the question of how these sequences can target survival genes through RNAi. We have generated new data to be published soon. These new data provide the first hint on how this might work. In the original manuscript, we showed in the highly DISE sensitive Drosha k.o. cells many more of the CD95L-derived reads are bound to AGO proteins when compared to wt cells. We have now performed an analysis of the total small RNAs in Drosha k.o. cells before they bind to AGO proteins. Overall, we find reads aligned to roughly to the same 22 regions in CD95L. However, overall reads in the cytosol tend to be longer than the ones bound to AGOs. In addition, we now have data on an RNA secondary structure prediction analysis for the CD95L ORF. According to this analysis the CD95L ORF RNA forms a tightly folded structure with many of the reads in the 22 regions juxtaposing each other in stem-like structures creating regions of significant complementarity. These may provide the duplexes needed to be processed and loaded into the RISC but this requires further work. We do not know yet what mRNA degrading system generated the small RNAs that are bound by AGO proteins. However, we see similar read locations when analyzing small CD95L derived RNAs in HeyA8 cells, both from total RNA as well as AGO bound (unpublished data) which could point at a general degradation mechanism. These new data will be part of the new manuscript we are now finalizing.